# Cytoplasmic Endonuclease G promotes nonalcoholic fatty liver disease via mTORC2-AKT-ACLY and endoplasmic reticulum stress

Wenjun Wang [1,2,8,9] ✉, Junyang Tan[1,2,8], Xiaomin Liu[1,2], Wenqi Guo [1,2], Mengmeng Li[1,2], Xinjie Liu[1,2], Yanyan Liu[1,2], Wenyu Dai [1,2], Liubing Hu[3], Yimin Wang[4], Qiuxia Lu[5], Wen Xing Lee[6], Hong-Wen Tang [6,7] & Qinghua Zhou [1,2,3,9] ✉

Endonuclease G (ENDOG), a nuclear-encoded mitochondrial intermembrane space protein, is well known to be translocated into the nucleus during apoptosis. Recent studies have shown that ENDOG might enter the mitochondrial matrix to regulate mitochondrial genome cleavage and replication. However, little is known about the role of ENDOG in the cytosol. Our previous work showed that cytoplasmic ENDOG competitively binds with 14-3-3γ, which released TSC2 to repress mTORC1 signaling and induce autophagy. Here, we demonstrate that cytoplasmic ENDOG could also release Rictor from 14-3-3γ to activate the mTORC2-AKT-ACLY axis, resulting in acetyl-CoA production. Importantly, we observe that ENDOG could translocate to the ER, bind with Bip, and release IRE1a/PERK to activate the endoplasmic reticulum stress response, promoting lipid synthesis. Taken together, we demonstrate that loss of ENDOG suppresses acetyl-CoA production and lipid synthesis, along with reducing endoplasmic reticulum stress, which eventually alleviates high-fat diet-induced nonalcoholic fatty liver disease in female mice.

Nonalcoholic fatty liver disease (NAFLD) has become the most common chronic disease (affecting approximately 25% of the worldwide population) due to the obesity pandemic[1]. Accumulation of lipids is the typical hallmark of NAFLD, which contributes to cellular stress, hepatic injury, and eventually the development of NAFLD[2]. Increased hepatocyte lipid content destroys ER homeostasis and initiates chronic ER stress, which could activate the unfolded protein response (UPR)[3]. In turn, the UPR plays roles in the pathogenesis of steatosis, including de novo lipogenesis, very-low-density-lipoprotein (VLDL) secretion, impaired autophagy, and apoptosis.

Lipid accumulation, ER stress, inflammation, and cell death are the combined hits for NAFLD.

Mitochondria are dynamic organelles for energy production, linking glucose and lipid metabolism through glycolysis, the Krebs cycle, and β-oxidation. Acetyl-CoA acts as a critical component in metabolism due to its intersection with many metabolic pathways and transformations, including the synthesis of ATP, ketone bodies, lipids, cholesterol, steroids, and protein acetylation modifications[4]. Acetyl-CoA is synthesized in mitochondria by oxidative decarboxylation of pyruvate, catabolism of some amino acids, β-oxidation of fatty acids,

[1]The Sixth Affiliated Hospital of Jinan University (Dongguan Eastern Central Hospital), Jinan University, Dongguan, Guangdong 523067, China. [2]The Biomedical Translational Research Institute, Health Science Center (School of Medicine), Jinan University, Guangzhou, Guangdong 510632, China. [3]The First Affiliated Hospital, Jinan University, Guangzhou, Guangdong 510632, China. [4]GeneMind Biosciences Company Limited, No. 116, Qingshuihe 1st Road, Luohu District, Shenzhen, Guangdong 518000, China. [5]School of Food and Biological Engineering, Chengdu University, Chengdu 610106, China. [6]Program in Cancer and Stem Cell Biology, Duke-NUS Medical School, 8 College Road, Singapore 169857, Singapore. [7]Division of Cellular & Molecular Research, Humphrey Oei Institute of Cancer Research, National Cancer Centre Singapore, Singapore 169610, Singapore. [8]These authors contributed equally: Wenjun Wang, Junyang Tan. [9]These authors jointly supervised this work: Wenjun Wang, Qinghua Zhou. ✉e-mail: wenjun-wang.jnu@foxmail.com; gene@email.jnu.edu.cn

and many other molecules[5]. Acetyl-CoA cannot be transported directly across the inner mitochondrial membrane to the cytosol, thus cytosolic generation of acetyl-CoA occurs mainly through the citrate shuttle pathway by the citrate carrier. Cytoplasmic citrate is converted back into acetyl-CoA by ATP citrate lyase (ACLY)[6]. Activation of ACLY is regulated by phosphorylation and acetylation. PI3K/AKT is the first identified pathway that phosphorylates ACLY on serine 455 and promotes its catalytic activity[7,8]. Recent research has demonstrated that DNA damage and mTORC2/AKT signaling also promote the activation of ACLY, which enhances lipogenesis[9,10].

Endonuclease G (ENDOG), a DNA/RNA nonspecific ββα-Me-finger nuclease, is an evolutionarily conserved mitochondrial protein involved in several biological functions[11]. ENDOG is encoded by a nuclear gene as a precursor protein with 297 amino acids (approximately 32 KD), and imported into the mitochondria by removing the mitochondrial targeting sequence (MTS) 1-48 amino acids to generate the mature form of ENDOG (approximately 27 KD)[12,13]. ENDOG is released from the mitochondrial intermembrane space and translocated into the nucleus to fragment chromosomal DNA during apoptosis[14]. In addition, ENDOG can enter the mitochondrial matrix to regulate mitochondrial genome cleavage and replication[15–17].

Recent studies have demonstrated that ENDOG is involved in multiple human diseases. ENDOG acts as a critical downstream executor of α-synuclein cytotoxicity during Parkinson's disease[18]. In HBV-associated hepatocellular carcinoma, the nuclear translocation of ENDOG causes genome instability in hepatocytes and promotes the development of hepatocellular carcinoma[19]. ENDOG is involved in cardiac hypertrophy and mitochondrial myopathy[17,20]. Loss of ENDOG promotes thermogenic gene expression and browning of white adipose tissue, as well as better glucose tolerance and fat mass in ENDOG knockout mice[21]. However, whether ENDOG regulates hepatic lipid metabolism and NAFLD remains unknown.

Our previous study reported that ENDOG degrades paternal mitochondrial DNA to eliminate paternal mitochondria upon fertilization[22]. Furthermore, we found that ENDOG promoted autophagy by suppressing the mTORC1 pathway and DNA damage[23]. When visualizing the autophagosome formation of ENDOG knockout cells and mouse livers by transmission electron microscopy (TEM), we also observed strikingly decreased lipid droplets, which drew our attention. In the present study, we investigated the regulatory role and mechanism of ENDOG in hepatic lipid metabolism. We demonstrated that ENDOG is released into the cytoplasm and competitively binds with 14-3-3γ to dissociate the Rictor protein, activating the mTORC2/ AKT/ACLY axis and eventually contributing to acetyl-CoA production and lipid accumulation. In addition, we surprisingly observed that cytoplasmic ENDOG is also translocated to the ER and then binds with Bip to activate ER stress, which promotes the expression of the lipid synthesis enzymes ACC and FAS. Collectively, we identified that ENDOG depletion alleviates HFD-mediated NAFLD by reducing acetyl-CoA production, de novo lipogenesis, and ER stress through translocation to the cytosol and ER.

## Results
### Loss of ENDOG represses lipid accumulation in hepatocytes
Our previous study showed that ENDOG knockout represses starvation-induced autophagy in mouse livers[23]. Serendipitously, we observed a dramatically reduced number of lipid droplets in the electron microscopy results, suggesting that ENDOG knockout may play an essential role in regulating hepatic lipid metabolism. To confirm the association between ENDOG and lipid metabolism, we detected lipid droplets in ENDOG knockout mice and hepatocytes. We found that, after 24 hours of starvation, the number but not the size of lipid droplets was reduced in male ENDOG knockout mouse (Endog[-/-]) livers compared to the livers of littermate control mice (Endog[+/-]) (Fig. 1a–c). The total triglyceride level was significantly reduced in starved ENDOG knockout mouse livers (Fig. 1d). Additionally, the BODIPY staining also showed that loss of

ENDOG reduced the lipid area in mouse livers after 24-hour starvation (Fig. 1e, f). In a hepatocytic cell line, we also found that knockout of ENDOG reduced lipid accumulation under normal conditions. ENDOG depletion dramatically repressed oleic acid treatment-induced lipid accumulation and triglyceride production (Fig. 1g–i). Additionally, we found that loss or knockdown of ENDOG significantly reduced the expression of PLIN2 (lipid droplet marker) in the normal or oleic acid treatment conditions (Fig. S1). Conversely, overexpression of ENDOG promoted lipid accumulation and triglyceride production upon oleic acid or control treatment (Fig. 1j–l). In C. elegans, we found that loss of CPS-6, the homolog of ENDOG, also repressed lipid accumulation under standard or egg yolk supplementation (high fat mimic) conditions (Fig. S2). These data suggest that loss of ENDOG represses lipid accumulation in vivo and in vitro.

### ENDOG promotes lipid accumulation through AKT-ACLY-mediated acetyl-CoA production
To identify pathways that potentially underlie ENDOG-mediated lipid accumulation, we performed RNA-Seq transcriptome analyses in wild-type and ENDOG overexpressing cells. The distribution of gene expression between wild-type and ENDOG overexpressing cells in the control and oleic acid treatment groups is represented by a volcano plot (Fig. S3a). Compared with the wild-type cells, we identified 551 and 619 significantly upregulated genes under control and oleic acid treatment, respectively, in ENDOG overexpressing cells. Conversely, we identified that 406 and 290 genes were downregulated in ENDOG overexpressing cells under control and oleic acid treatment, respectively (Fig. S3b). We focused on the upregulated genes in ENDOG overexpressing cells, and Gene Ontology (GO) analyses showed that ENDOG might regulate the PI3K-AKT pathway (Fig. S3c). ENDOG indeed promoted the phosphorylation of AKT at Ser 473 and Thr 308, as well as its downstream target ACLY. Conversely, knockout of ENDOG repressed the phosphorylation of AKT and ACLY (Figs. 2a, b and S4a, b). PTEN and PI3K are two essential regulators of AKT phosphorylation at Thr308. We found that neither overexpression nor knockout of ENDOG affected the expression or phosphorylation of PTEN or the expression of PI3K (Fig. S3d). However, we found that ENDOG binds with PI3K, and their binding was enhanced following oleic acid treatment (Fig. S5), suggesting that ENDOG might promote the phosphorylation of AKT at Thr308 through binding with PI3K.

ENDOG activates the AKT-ACLY axis, and ACLY is a critical enzyme that converts citrate and acetate into acetyl-CoA, which is the main source for lipid biosynthesis[6,24,25]. Thus, we hypothesized that the repressed lipid synthesis in ENDOG knockout cells might be due to the deficiency of acetyl-CoA. As predicted, we detected a significantly decreased level of acetyl-CoA in ENDOG knockout cells and mouse livers. However, overexpression of ENDOG increased acetyl-CoA in hepatocytes (Fig. 2c, d). Besides citrate, acetate is another pool for acetyl-CoA synthesis. ALDH2 and AceCS1 are critical enzymes for acetaldehyde converting to acetate and acetate to acetyl-CoA, respectively. We found ENDOG did not regulate the expression of ALDH2 and AceCS1 (Fig. S6). Citrate provides acetyl-CoA for lipid biosynthesis and promotes lipid synthesis by activating ACC[26], the first enzyme for fatty acid synthesis. We found that extra citrate supplementation increased lipid accumulation in wild-type cells but could not restore lipid droplet accumulation and triglyceride concentrations in ENDOG knockout cells (Fig. 2e–g). However, acetyl-CoA supplementation restored the decreased triglyceride content and lipid droplet accumulation in ENDOG knockout cells (Fig. 2h–j). Taken together, these data demonstrate that ENDOG promotes lipid synthesis by activating AKT/ACLY-mediated acetyl-CoA production.

To validate that ENDOG-mediated lipid accumulation is dependent on the activation of AKT-ACLY, we performed experiments with the PI3K inhibitor LY294002 or continuously activated AKT. LY294002 treatment efficiently abolished ENDOG-mediated AKT/ACLY activation,

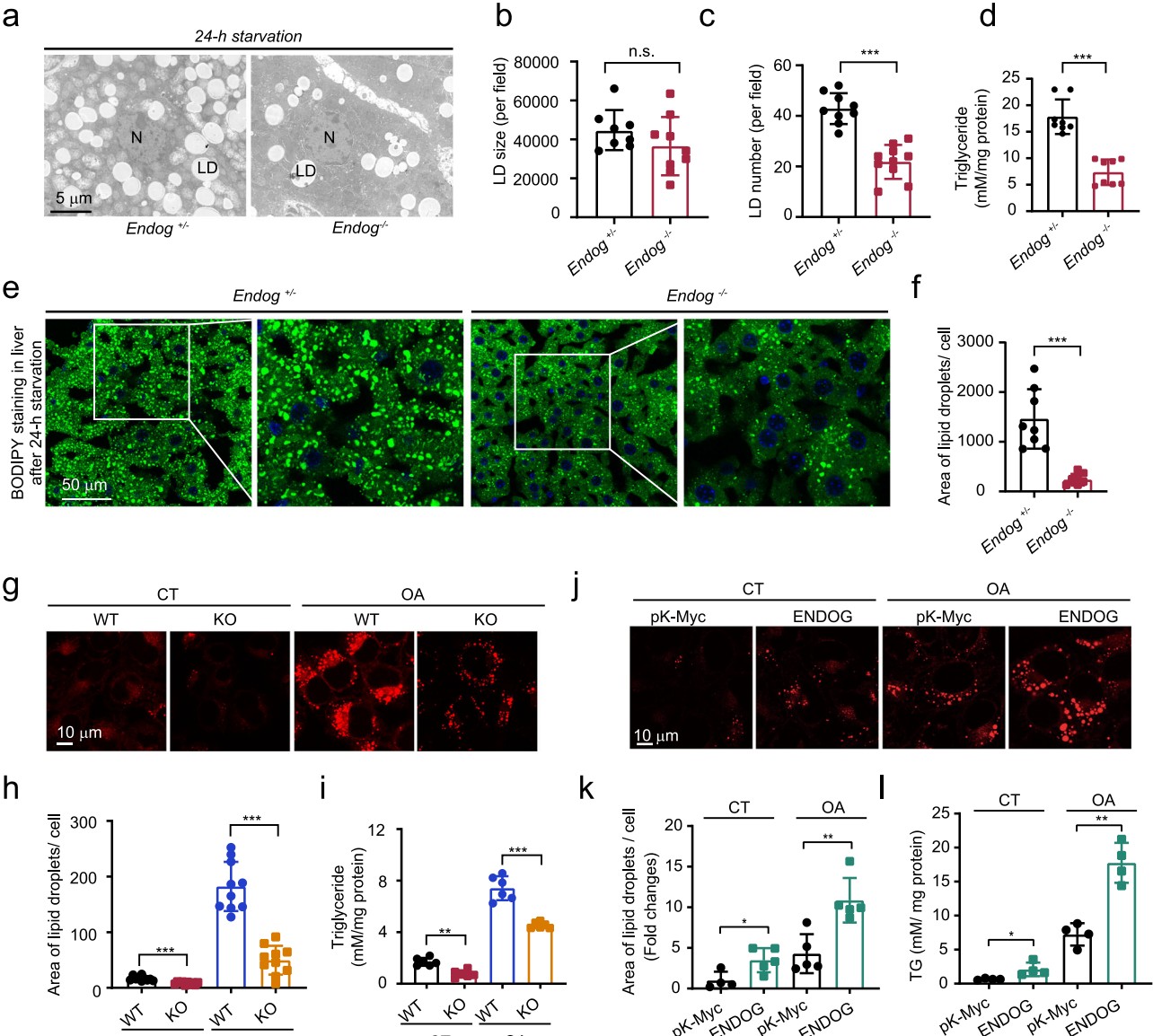

**Fig. 1 | Loss of ENDOG represses lipid accumulation in mouse livers and hepatocytes. a–c** Representative electron microscopic images and quantification of lipid size and number in *Endog⁺/⁻* or *Endog⁻/⁻* mouse livers after 24 hours of starvation. N: Nuclear; LD: lipid droplet; *n* = 8mice. **d** Measurement of total triglycerides in mouse livers after 24 hours of starvation. *n* = 8 mice. **e, f** Representative images of BODIPY staining and the quantitative results of lipid area in liver tissue of mice starved for 24 hours. *n* = 8 mice. **g–i** Representative images of Nile red staining, the quantitative results of lipid area, and measurement of triglycerides in WT and ENDOG KO HepG2 cells after treatment of 200 µM oleic acid for 24 h. *n* = 10 independent samples for Nile red staining and 6 for triglyceride measurement. **j–l** Representative images of Nile red staining, the quantitative results of lipid area, and measurement of triglycerides in control and ENDOG overexpressing HepG2 cells following the treatment of 200 µM oleic acid for 24 h. pK-Myc and ENDOG overexpressing plasmids were transfected for 48 h. *n* = 4 independent samples for Nile red staining and 5 for triglyceride measurement. Statistical significance was determined by unpaired Student's t-test (two-tailed) in (**b**, **c**, **d**, **f**, **h**, **i**, **k**, **l**); error bars are mean ± SD. Source data and exact *P* values are provided in a Source data file. *P < 0.05; **P < 0.01; ***P < 0.001; n.s.: no significance.

lipid accumulation and triglyceride production (Figs. 2k–n, and S4c). Notably, reintroducing continuously activated AKT (myr-AKT) increased the phosphorylation of AKT and ACLY in both wild-type and ENDOG knockout cells (Fig. 2o). The decreased p-ACLY level in ENDOG knockout cells was restored to a level similar to that in wild-type cells after overexpression of myr-AKT (Fig. 2o). Next, ACLY, myr-AKT, or ACLY/myr-AKT were overexpressed in ENDOG knockout cells. Interestingly, we found that overexpression of myr-AKT or myr-AKT/ACLY together, but not ACLY alone, could increase lipid droplet accumulation and triglyceride production in ENDOG knockout cells under oleic acid treatment (Figs. 2p–r and S7a, b). Insulin and insulin receptor signaling are critical regulators of PI3K-AKT activation. Consistently, insulin treatment also restored the activation of AKT-ACLY in ENDOG knockout

cells (Fig. S7c, d). In addition, insulin supplementation promoted lipid accumulation in ENDOG knockout cells under oleic acid treatment (Fig. S7e–h). These data suggest that AKT-mediated activation of ACLY is crucial for ENDOG-promoted lipid synthesis. Furthermore, we knocked down ACLY in ENDOG overexpressing cells and found that knockdown of ACLY suppressed ENDOG overexpression-mediated lipid accumulation under control and oleic acid treatment conditions (Fig. S8). Together, ENDOG promotes lipid accumulation via activated AKT-ACLY axis-mediated acetyl-CoA synthesis.

**Loss of ENDOG represses HFD-induced liver lipid accumulation**
To investigate whether ENDOG regulates liver lipid accumulation in a high-fat diet model, ENDOG systemic and liver-specific knockout male

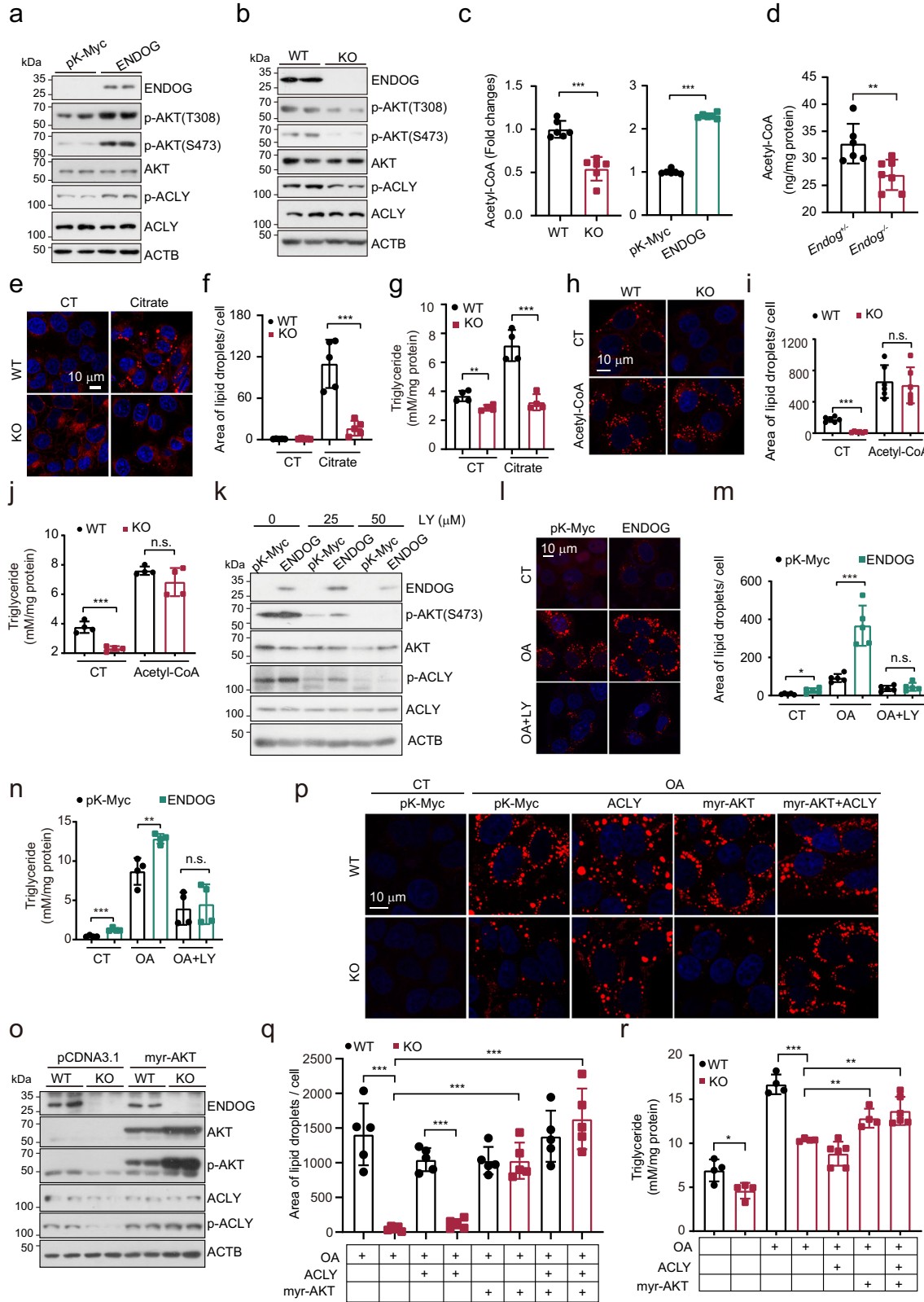

and female mice were fed a HFD for 16 weeks. We observed a slower body weight increase during high-fat diet feeding in ENDOG knockout female mice compared to that of the control group (Fig. 3a, b). While the loss of ENDOG did not affect the body weight, nonfasting blood glucose (NFBG), and organ weights in male mice after a high-fat diet (Figs. S9 and S10c, d). The anatomical results showed that, compared to the littermate control mice, the weight and size of eWAT, iWAT, BAT,

and liver weight in ENDOG knockout mice were reduced (Fig. 3c, d). However, in male and female mice, loss of ENDOG did not affect non-fasting blood glucose (NFBG) in normal and high-fat diet conditions (Figs. 3d and S10a–f). The fasting blood glucose (FBG) also had no changes in male and female mice after the high-fat diet (Fig. S10g, h). These data indicated that loss of ENDOG may not affect the blood glucose under normal or high-fat diet conditions. By RNA-seq followed

**Fig. 2 | ENDOG promotes lipid synthesis via the AKT-ACLY axis. a**, **b** Western blot analyses of activation of AKT-ACLY in ENDOG overexpressing and knockout HepG2. pK-Myc and ENDOG overexpressing plasmids were transfected for 48 h. **c** Measurement of acetyl-CoA in ENDOG overexpressing / knockout HepG2. *n* = 6 biologically independent samples each group. **d** Measurement of acetyl-CoA in *Endog*[+/-] and *Endog*[-/-] mice livers. *n* = 6 mice in *Endog*[+/-] group and 7 mice in *Endog*[-/-] group. **e**–**g** Representative images of Nile red, the quantitative results of lipid area, and measurement of triglycerides in wild-type and ENDOG knockout HepG2 after the supplementation of 10 mM citrate for 24 h. *n* = 5 biologically independent samples for Nile red staining and 4 for triglyceride measurement. **h**–**j** Representative images of Nile red, the quantitative results of lipid area, and measurement of triglycerides in wild-type and ENDOG knockout HepG2 after supplementing 50 μM acetyl-CoA for 24 h. *n* = 6 biologically independent samples for Nile red staining and 4 for triglyceride measurement. **k** Western blot analyses of AKT-ACLY signaling in ENDOG overexpressing HepG2 following 50 or 100 μM

LY294002 treatment for 24 h. pK-Myc and ENDOG overexpressed plasmids were transfected for 48 h. **l**–**n** Representative images of Nile red, the quantitative results of lipid area and measurement of ENDOG overexpressing HepG2 following the treatment of 200 μM oleic acid and 50 μM LY294002 for 24 h. *n* = 5 biologically independent samples for Nile red staining and 4 for triglyceride measurement. **o** Western blot analyses of AKT-ACLY signaling in wild-type and ENDOG knockout HepG2 after transfection with myr-AKT plasmid for 48 h. **p**–**r** Representative images of Nile red, the quantitative results of lipid area, and measurement of triglycerides in the indicated groups. Cells were transfected with plasmids for 24 h and then treated with 200 μM oleic acid for another 24 h. *n* = 5 biologically independent samples for Nile red staining and 4-6 for triglyceride measurement. Statistical significance was determined by unpaired Student's t-test (two-tailed) in (**c**, **d**, **f**, **g**, **i**, **j**, **m**, **n**, **q**, **r**); error bars are mean ± SD. Source data and exact *P* values are provided in a Source data file. *$P < 0.05$; **$P < 0.01$; ***$P < 0.001$; n.s.: no significance.

by GO pathway analyses, we found that several lipid metabolism processes, including fatty acid metabolism, lipid biosynthetic process, and acyl-CoA hydrolase activity, were downregulated in ENDOG knockout female mice livers (Fig. S11a). Similarly, gene set enrichment analyses (GSEA) of the RNA-seq results showed that fatty acid metabolism (NES = −1.32, p-value = 0.031), and adipogenesis (NES = −1.38, p-value = 0.008) were enriched in littermate control female mice livers (Fig. S11b), suggesting that ENDOG knockout suppressed lipid metabolism. H&E staining of liver sections showed that loss of ENDOG ameliorated HFD-induced hepatic steatosis (Fig. 3e, f). The total triglycerides in the liver and free fatty acids in serum were also reduced in ENDOG knockout female mice after HFD chow feeding (Fig. 3g, h). To verify that the loss of ENDOG repressed liver lipid accumulation, we generated liver-specific knockout mice (*Endog* [LKO]) and control mice (*Endog* [flox]), and then fed them with HFD. Liver-specific loss of ENDOG repressed body weight, white fat tissue and liver weight, but did not affect NFBG and brown fat tissue weight (Figs. 3i–k and S10e, f) in female mice. HFD-induced liver steatosis was also alleviated in *Endog* [LKO] mouse livers compared to *Endog* [flox] mouse livers (Fig. 3l, m). The total triglycerides in the liver and free fatty acids in serum were also reduced in *Endog* [LKO] mouse livers (Fig. 3n, o). Consistent with the ENDOG knockout cell results, we detected decreased phosphorylation of AKT and ACLY in both systemic and liver-specific ENDOG knockout female mice livers under HFD condition (Fig. 3p–s). Our findings suggest that loss of ENDOG alleviates HFD-induced NAFLD by blocking the AKT-ACLY axis. We also performed tolerance tests in ENDOG knockout male and female mice. We found that loss of ENDOG did not affect glucose tolerance and insulin sensitivity in male mice after a high-fat diet (Fig. S12a–d). While in female mice, loss of ENDOG did not influence glucose tolerance but increased insulin sensitivity under the high-fat diet conditions (Fig. S12e–h). Next, we measured the serum insulin in female mice under normal chow, starvation, and high-fat diet conditions. We found that loss of ENDOG had no changes in serum insulin under these conditions (Fig. S12i, j). These data indicated that loss of ENDOG could not affect the glucose tolerance and insulin sensitivity after the HFD in male mice but slightly increased insulin sensitivity in the female mice.

## ENDOG activates the mTORC2-AKT-ACLY axis by competitively binding with 14-3-3γ

Next, we determined how ENDOG promotes the activation of AKT-ACLY. mTORC2 has been reported to activate the AKT-ACLY axis, promoing lipogenesis and adipogenesis in adipocytes[10]. Jeon et al. reported that the binding between 14-3-3 and Rictor (an essential component of mTORC2) repressed the activity of mTORC2, which blocked the phosphorylation of AKT at Ser473[27]. Our previous work demonstrated that ENDOG represses mTORC1 activity by interacting with 14-3-3γ, which releases TSC2, the critical negative regulator of mTORC1[23]. We detected that ENDOG knockout decreased the phosphorylation of mTORC2 downstream targets, including AKT and SGK1

(Figs. 2a, b and S13). We hypothesized that the binding between ENDOG and 14-3-3γ could also regulate mTORC2 signaling. Indeed, we found that overexpression of ENDOG repressed the interaction between 14-3-3γ and Rictor (Fig. 4a, b). The endogenous Co-IP results showed that the loss of ENDOG enhanced the binding between 14-3-3γ and Rictor (Fig. 4c). By subcellular fraction isolation, we observed that oleic acid treatment promoted the release of ENDOG from mitochondria to the cytoplasm (Fig. 4d, e). Costaining of Tim23 (a mitochondrial inner membrane protein, indicating mitochondria) and ENDOG showed that oleic acid treatment promoted the cytoplasmic translocation of ENDOG (Fig. 4f). Importantly, oleic acid treatment enhanced the binding between ENDOG and 14-3-3γ, and attenuated the interaction of 14-3-3γ and Rictor (Fig. 4g). These data indicate the competitive binding between ENDOG and 14-3-3γ released Rictor, eventually activating the mTORC2 signaling pathway. Next, we investigated whether 14-3-3γ could affect ENDOG-mediated AKT-ACLY activation and lipid accumulation. The results showed that 14-3-3γ overexpression inhibited the ENDOG-mediated activation of AKT-ACLY (Fig. 4h, i), lipid accumulation and total triglyceride levels (Fig. 4j–l). These data demonstrate that ENDOG is released into the cytoplasm under oleic acid treatment and then competitively binds with 14-3-3γ to release Rictor, eventually activating the mTORC2-AKT-ACLY axis and resulting in lipid accumulation.

## ENDOG promotes the ER stress
The loss of ENDOG reduced acetyl-CoA, the lipid synthesis source (Fig. 2c, d). We wondered whether ENDOG regulated lipid metabolism, including lipid synthesis, uptake, oxidation, and export. ENDOG depletion repressed the transcription of the lipid synthesis-associated genes *ACC1*, *ACC2*, and *FAS* but did not affect the transcription of lipid uptake, oxidation, and export-related genes (Fig. 5a). Furthermore, ENDOG promoted the expression of SCD1 (an enzyme of monounsaturated fatty acid) but did not affect the expression of DGAT1 and DGAT2, which catalyze the final step of triglyceride synthesis (Fig. S14). ENDOG also did not affect the expression of CPT1 and CPT2 (two critical enzymes in fatty acid β-oxidation) both in vitro and in vivo (Fig. S15). Moreover, knockout of ENDOG repressed the protein levels of FAS and ACC, while ENDOG overexpression promoted their expression (Figs. 5b, c and S16a). By GO-CC analyses of the RNA-seq data, we found that ER- associated gene sets, including the lumenal side of the endoplasmic reticulum membrane (NES = −1.86, *p* value = 0.000), endoplasmic reticulum protein-containing complex (NES = −1.45, *p* value = 0.000), and mitochondria-associated endoplasmic reticulum membrane (NES = −1.62, *p* value = 0.017) were downregulated in ENDOG overexpressing cells under oleic acid treatment (Fig. 5d), suggesting that ENDOG may regulate ER function.

ER stress is another risk factor for NAFLD development based on the multiple-hit pathogenesis of NAFLD[28]. ER stress induces lipid accumulation, insulin resistance, inflammation and cell death, which

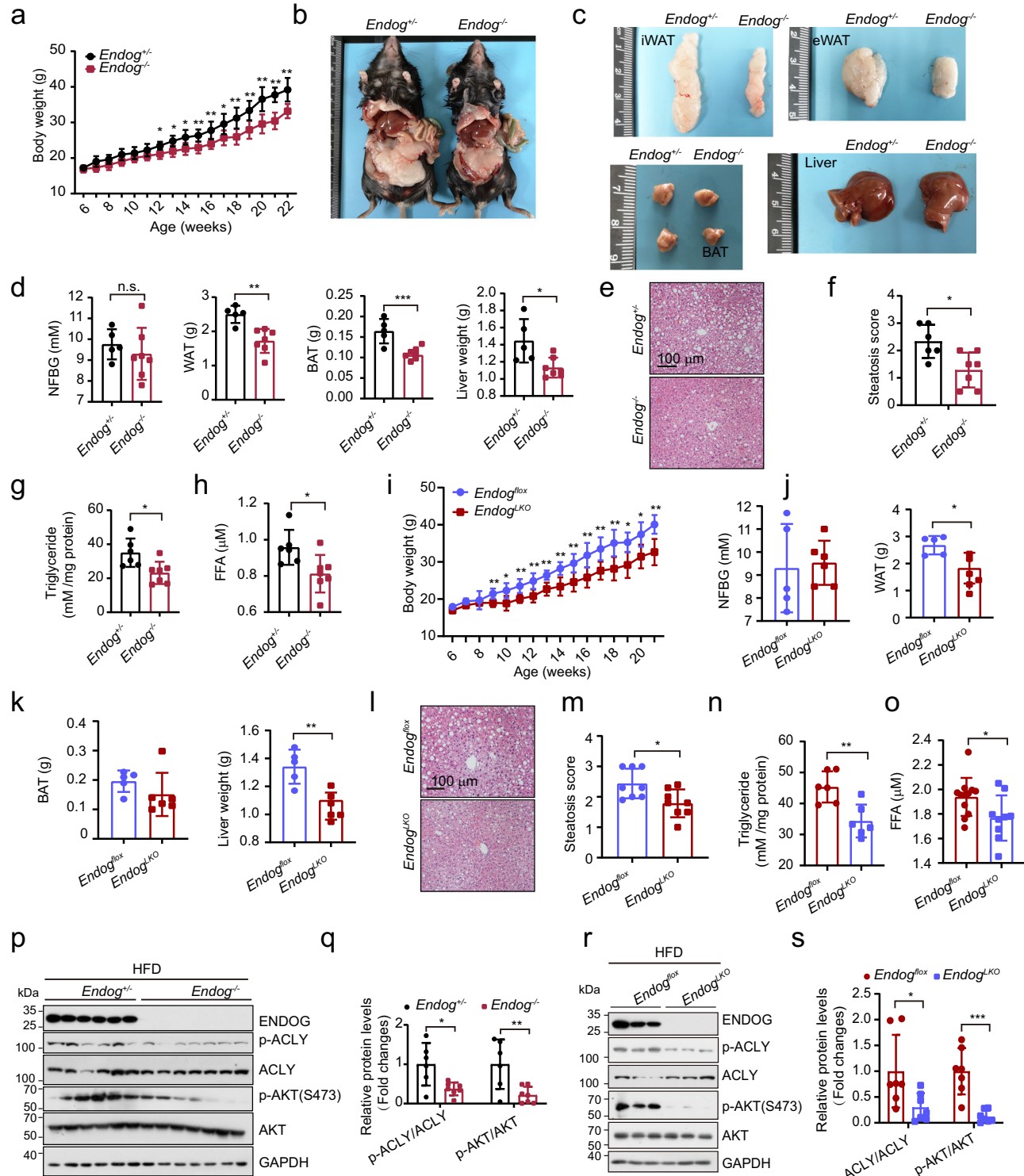

**Fig. 3 | Loss of ENDOG represses HFD-induced lipid accumulation by suppressing the AKT-ACLY axis. a** Body weight of HFD-fed female mice at different ages. *n* = 6 mice in *Endog⁺/⁻* group and 7 mice in *Endog⁻/⁻* group. **b, c** Gross images of inguinal white adipose (iWAT), epididymal white adipose tissue (eWAT), brown adipose tissue (BAT) and liver of the HFD-fed female mice. **d** Nonfasting blood glucose (NFBG), the weight of WAT (iWAT and eWAT), BAT, and liver in HFD-fed female mice. *n* = 5 mice in *Endog⁺/⁻* group and 7 mice in *Endog⁻/⁻* group. **e** Liver H&E staining in HFD female mice. **f** Steatosis score of HFD female mice liver. *n* = 6 mice in *Endog⁺/⁻* group and 7 mice in *Endog⁻/⁻* group. **g, h** Measurement of triglycerides in the liver and free fatty acid in serum. *n* = 6 mice in *Endog⁺/⁻* group and 7 mice in *Endog⁻/⁻* group. **i–k** Body weight, NFBG, the weight of WAT, BAT, and liver in ENDOG liver-specific knockout female mice. *n* = 5 mice in *Endog⁺/⁻* group and 6 mice in *Endog⁻/⁻* group. **l, m** Liver H&E staining and steatosis score. *n* = 8 mice each group. **n** Measurement of triglycerides in the liver. *n* = 6 mice each group. **o** Measurement of free fatty acid in serum. *n* = 12 mice in *Endog⁺/⁻* group and 9 mice in *Endog⁻/⁻* group. **p, q** Western blot analyses of AKT-ACLY signaling in *Endog⁺/⁻ and Endog⁻/⁻* HFD mice livers. *n* = 6 mice in *Endog⁺/⁻* group and 7 mice in *Endog⁻/⁻* group. **r, s** Western blot analyses of AKT-ACLY signaling in *Endog^flox and Endog^LKO* HFD mice livers. *n* = 6 mice in *Endog⁺/⁻* group and 7 mice in *Endog⁻/⁻* group. Statistical significance was determined by unpaired Student's t-test (two-tailed) in (**a, d, f, g, h, i, j, k, m, n, o, q, s**); error bars are mean ± SD. Source data and exact *P* values are provided in a Source data file. *P < 0.05; **P < 0.01; ***P < 0.001; n.s.: no significance.

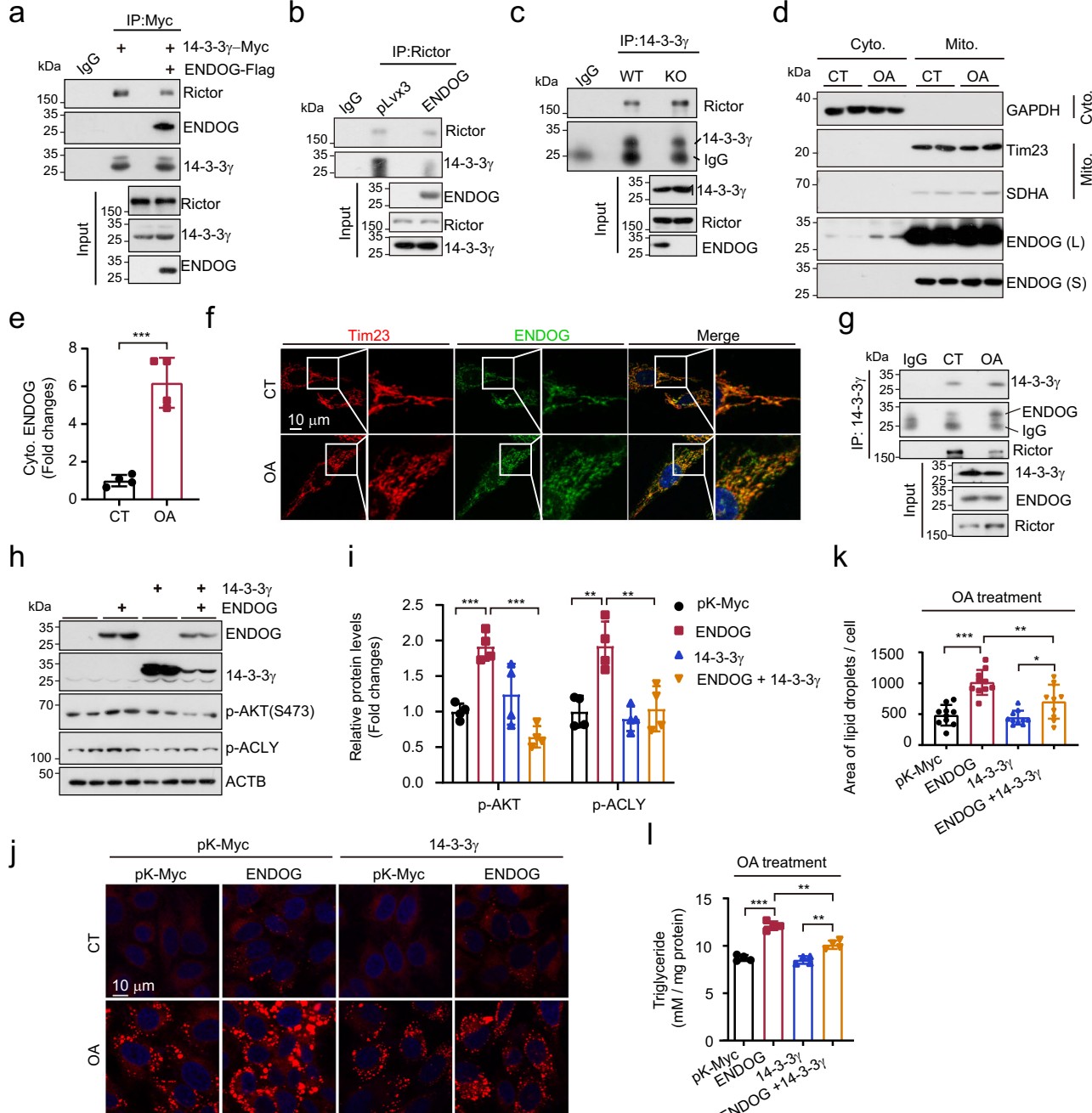

**Fig. 4 | ENDOG activates the mTORC2-AKT-ACLY axis by competitively binding with 14-3-3γ. a** Co-IP analyses. HepG2 cells were cotransfected with 14-3-3γ-Myc and ENDOG-Flag/ pLVX3-Flag plasmids for 48 h. **b** Co-IP analyses. HepG2 cells were transfected with ENDOG-Flag or pLVX3-Flag plasmids for 48 h. **c** Endogenous Co-IP analyses in wild-type and ENDOG knockout HepG2 cells. **d, e** ENDOG releases to the cytoplasm in HepG2 after being treated with 200 μM oleic acid for 24 h. Cyto, cytoplasm; Mito, mitochondria. *n* = 4 biologically independent samples. **f** Representative images of costaining of Tim23 (mitochondrial inner membrane protein) and ENDOG in HepG2 after being treated with 200 μM oleic acid for 24 h. **g** Endogenous Co-IP analyses in wild-type HepG2 cells after being treated with 200 μM oleic acid for 24 h. **h, i** Western blot and quantitative results of AKT-ACLY

signaling after transfection with ENDOG and 14-3-3γ / ENDOG plasmids together for 48 h. *n* = 4 biologically independent samples. **j–l** Representative images of Nile red, the quantitative results of lipid area and measurement of triglycerides after transfection with ENDOG and 14-3-3γ / ENDOG plasmids together for 24 h and then treated with 200 μM oleic acid for another 24 h. *n* = 10 biologically independent samples for Nile red staining and 4 for triglyceride measurement. Three experiments were repeated independently with similar results in (**a, b, c, f, g**). Statistical significance was determined by unpaired Student's t-test (two-tailed) in (**i, k, l**); error bars are mean ± SD. Source data and exact *P* values are provided in a Source data file. *\*P* < 0.05; *\*\*P* < 0.01; *\*\*\*P* < 0.001.

promote the pathogenesis of NAFLD[29]. ER stress promotes lipid synthesis through the maturation of SREBP1, a transcription factor that activates several genes, including *ACC1*, *ACC2*, and *FAS*[30]. Thus, we propose that ENDOG promotes NAFLD by enhancing ER stress and lipid synthesis. As expected, we found that loss of ENDOG repressed

the expression of two ER stress sensors, IRE1a and PERK (Fig. 5e). Treatment with ER stress inducers (tunicamycin and thapsigargin) activated ER stress in wild-type cells. However, ER stress in ENDOG knockout cells was significantly blocked (Figs. 5e and S16b). Consistently, SREBP1 expression and maturation, as well as the expression

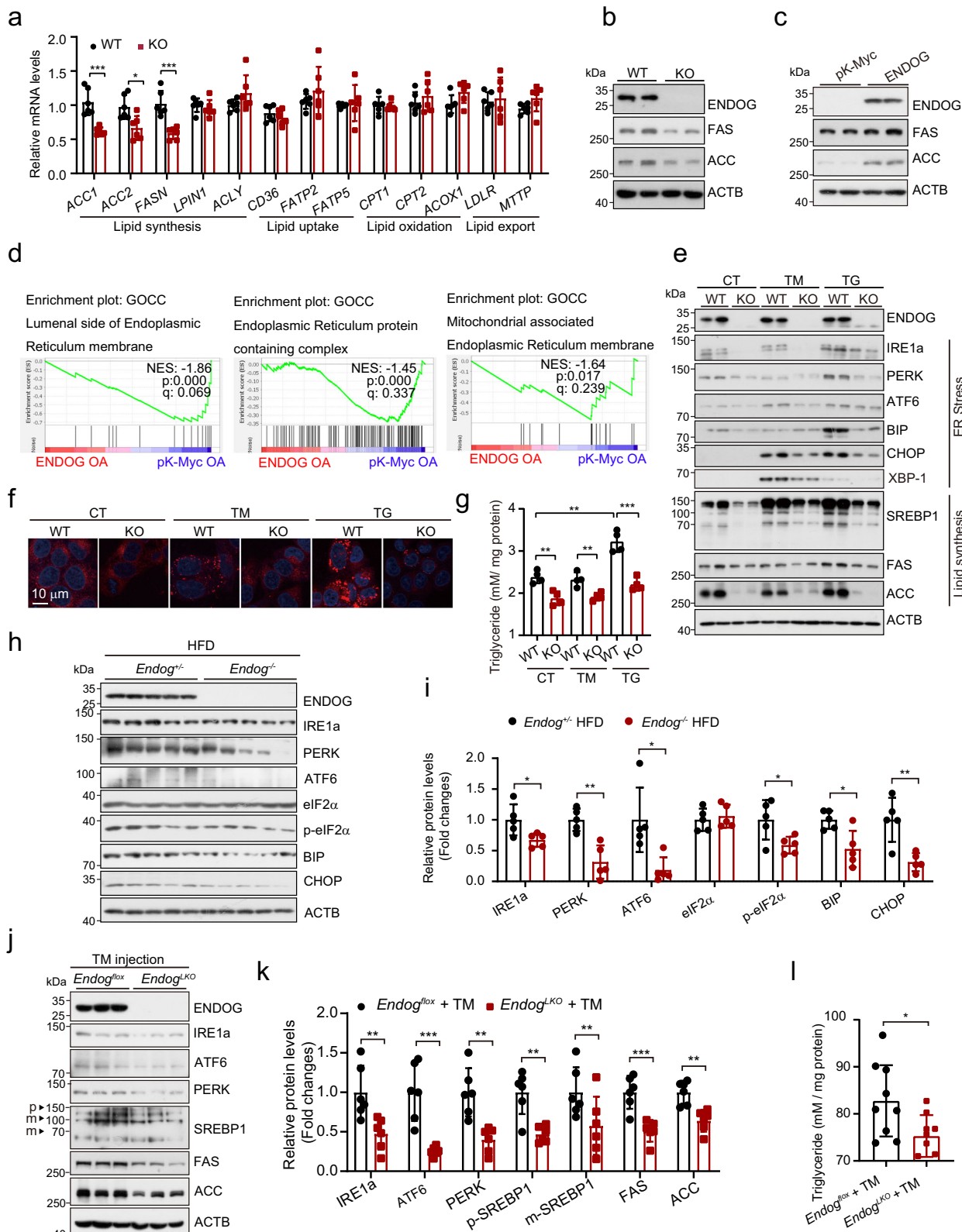

of ACC and FAS were repressed in ENDOG knockout cells under control or ER stress-induced conditions (Figs. 5e and S16b). Moreover, ENDOG depletion repressed ER stress induced lipid droplet accumulation and triglyceride synthesis when cells were treated with thapsigargin (Fig. 5f, g). In addition, the IRE1a-XBP1 branch has been reported to promote triglyceride synthesis by increasing the expression of Lipins[31]. We identified that ENDOG knockout represses the expression

of IRE1a, the splicing of XBP1, and the expression of Lipin1 (Fig. S17). Besides the SREBP1, there are other transcription factors regulated by ER stress, which also play critical roles in lipid metabolism, such as PPARγ, ChREBP1, FABP4, CRTC2, and CREBH. However, we found that only SREBP1 was reduced in response to loss of ENDOG (Figs. 5e and S18). These data suggested that SREBP1 may be the predominant transcriptional factor in ENDOG-induced lipid synthesis.

**Fig. 5 | ENDOG promotes lipid synthesis through the activation of ER stress.**
**a** qPCR results of lipid metabolism genes in wild-type and ENDOG knockout HepG2 cells. $n = 4$ biologically independent samples. **b**, **c** Western blot analyses of lipid synthesis proteins in HepG2. WT, wild-type, KO: ENDOG knockout; pK-Myc or ENDOG plasmids were transfected for 48 h. **d** GSEA shows endoplasmic reticulum-related gene sets are rich in ENDOG overexpressed HepG2 cells. **e** Western blot analyses of ER stress-related proteins and lipid synthesis proteins in wild-type and ENDOG knockout HepG2 cells following treatment with 2 μg/ml tunicamycin (TG) and 1 μM thapsigargin (TG) for 24 h. **f**, **g** Representative images of Nile red and measurement of triglycerides. Wild-type and ENDOG knockout HepG2 cells following treatment with 2 μg/ml tunicamycin (TM) and 1 μM thapsigargin (TG) for 24 h. $n = 4$ biologically independent samples. **h**, **i** Western blots analyses of ER stress-related proteins in systemic and liver-specific ENDOG knockout female mice livers after the HFD. $n = 5$ mice each group. **j**, **k** Western blot analyses of ER stress-related proteins and lipid synthesis proteins in liver-specific ENDOG knockout female mice livers. Mice were fasting for 12 h and then intraperitoneally injected with tunicamycin (TM) 2 mg/kg body weight for 24 h. $n = 6$ mice each group. **l** Measurement of triglycerides in liver. Female mice were fasting for 12 h and then intraperitoneally injected with 2 mg/kg body weight for 24 h. $n = 9$ mice each group. Three experiments were repeated independently with similar results in (**b**, **c**, **e**). Statistical significance was determined by unpaired Student's t-test (two-tailed) in (**a**, **i**, **k**, **l**); error bars are mean ± SD. Source data and exact $P$ values are provided in a Source data file. *$P < 0.05$; **$P < 0.01$; ***$P < 0.001$.

Furthermore, ENDOG deletion repressed the expression of ER stress sensors and UPR signaling in HFD mice livers (Fig. 5h, i). Additionally, ENDOG knockout repressed the expression of ER stress sensors and SREBP1 after tunicamycin injection. Similarly, the expression of FAS, ACC, and triglyceride levels were reduced in liver-specific ENDOG knockout mice livers upon tunicamycin treatment (Fig. 5j–l). Additionally, we knockdowned of ENDOG in other hepatocytes and found that deficiency of ENDOG repressed the expression of FAS, ACC, and Lipin1, as well as the activation of AKT-ACLY (Fig. S19). These data suggest that ENDOG promotes lipid synthesis via the activation of ER stress, which promotes FAS, ACC, and Lipin1 expression.

### ENDOG activates ER stress by translocating to the ER and binding with Bip

Next, we aimed to determine how ENDOG regulates ER stress. By reanalyzing our ENDOG IP-Mass data[23], we found that ENDOG interacted with a few chaperone proteins and ER resident or mitochondria-associated ER membrane (MAM) proteins, including Bip, Grp75, and Calnexin (Figs. 6a and S20). Our Co-IP results confirmed that ENDOG binds with Bip and Grp75 (Figs. 6b and S21a, b). By costaining of ENDOG and Grp75, we observed their colocalization in the normal condition, and their colocalization was enhanced after oleic acid or ER stress inducer treatment (Figs. 6c and S22a). Profoundly, we found that oleic acid treatment increased the colocalization of ENDOG with ER resident proteins, including Calnexin and Bip (Fig. 6d, e). ER stress inducers also increased the colocalization of ENDOG and Calnexin (Fig. S22b). Consistently, using the in situ PLA (proximity ligation assay), we also observed the endogenous interaction of ENDOG with Bip, which was increased by oleic acid treatment (Fig. 6f, g).

GRP75 is an essential tether protein facilitating MAM formation through the IP3R-GRP75-VDAC1 complex[32]. BIP(Grp78) is mainly found in the ER lumen due to the presence of an ER signaling peptide[33]. Therefore, we hypothesized that ENDOG is released from mitochondria and might translocate to the MAM or ER lumen during stress. To verify that ENDOG could translocate to the ER, we isolated ER and mitochondrial fractions by the differential sucrose gradient isolation assay. We detected mature ENDOG in the ER fraction, and oleic acid increased the level of ER localized ENDOG (Fig. 6h). Consistently, we found that oleic acid promoted ENDOG levels in the ER enrichment by using the ER Enrichment Kit (ER-036, Invent Biotechnologies) (Fig. 6i). These data demonstrate that stress caused the translocation of ENDOG to the ER. Bip is a central regulator of ER stress, which binds with ER stress sensors (IRE1a, PERK and ATF6) and blocks them to activate the UPR[34]. We examined whether the binding between ENDOG and Bip influences the interaction of Bip and ER stress sensors. Indeed, overexpression of ENDOG repressed the interaction of Bip with PERK and IRE1a (Fig. 6j). The interactions of Bip with PERK and IRE1a were enhanced in the absence of ENDOG (Fig. 6k). Furthermore, the binding between ENDOG and Bip was increased, while the binding between Bip and IRE1a / PERK was reduced under oleic acid treatment (Figs. 6l and S21c).

Next, we overexpressed Bip in the presence of ENDOG. We found that the ENDOG-mediated increase in IRE1a, PERK and ATF6 were abolished after overexpression of Bip (Fig. S23a, b). ENDOG overexpression induced SREBP1 maturation, and the expression of FAS and ACC was reduced when Bip was overexpressed (Fig. S23a, b). ENDOG-induced lipid accumulation was repressed after reintroduction of Bip under control conditions, but it did not change under oleic acid treatment (Fig. S23c). Take together, our findings demonstrate that under oleic acid treatment, ENDOG is released from mitochondria, and translocated to the ER, and then competitively binds with Bip, followed by activation of ER stress by dissociating IRE1a and PERK, resulting in enhanced expression and maturation of SREBP1 and transcription of FAS/ACC, which contribute to lipid synthesis and accumulation.

### Inhibiting the release of ENDOG represses oleic acid-induced lipid accumulation

Voltage-dependent anion channels (VDACs), which are mitochondrial outer membrane-located proteins, regulate the flow of mitochondrial metabolites and proteins[35]. The VDAC oligomerization inhibitor VBIT represses the release of mtDNA and cytochrome c from mitochondria[36,37]. Previous work reported that blocking VDAC with the VDAC antibody prevented ENDOG release[36]. We wondered whether inhibition of VDAC could block the release of ENDOG in our system. By costaining ENDOG and the mitochondrial matrix protein HSP60, we found that pre-treatment of VDAC inhibitor VBIT-12 repressed the release of ENDOG from mitochondria. (Fig. 7a). Subcellular fraction isolation also showed that VBIT-12 reduced the oleic acid-induced release of ENDOG (Fig. 7b). Moreover, oleic acid-induced ER stress, expression of FAS, ACC and activation of AKT-ACLY were repressed by pretreatment with VBIT-12 (Fig. 7c, d). Consistently, VBIT-12 partially reduced oleic acid-induced lipid accumulation (Fig. 7e–g). These data demonstrate that inhibiting ENDOG release from mitochondria can reduce lipid accumulation by blocking AKT-ACLY activation and FAS/ACC expression.

## Discussion

In summary, we revealed that ENDOG is released from mitochondria under a high-fat diet or ER stress inducer treatments. The cytoplasm-located ENDOG competitively binds with 14-3-3γ, which dissociates Rictor and activates the mTORC2-AKT-ACLY axis, resulting in acetyl-CoA production. Additionally, ENDOG transports to the endoplasmic reticulum and interacts with Bip, which releases IRE1a and PERK, followed by the activation of ER stress. Activated ER stress promotes the expression and maturation of SREBP1 and the transcription of FAS and ACC. ENDOG knockout reduces acetyl-CoA production, lipid synthesis, and ER stress, eventually alleviating HFD-induced NAFLD. (Fig. 7h).

ENDOG is mainly located in the mitochondrial intermembrane[38] and transports into the nucleus during apoptosis[14]. ENDOG has been reported to be released from mitochondria under specific stresses, such as infection, genotoxicity, and starvation[19,23,39]. In this study, we found that ENDOG is released from mitochondria under oleic acid treatment. Our previous study demonstrated that starvation caused

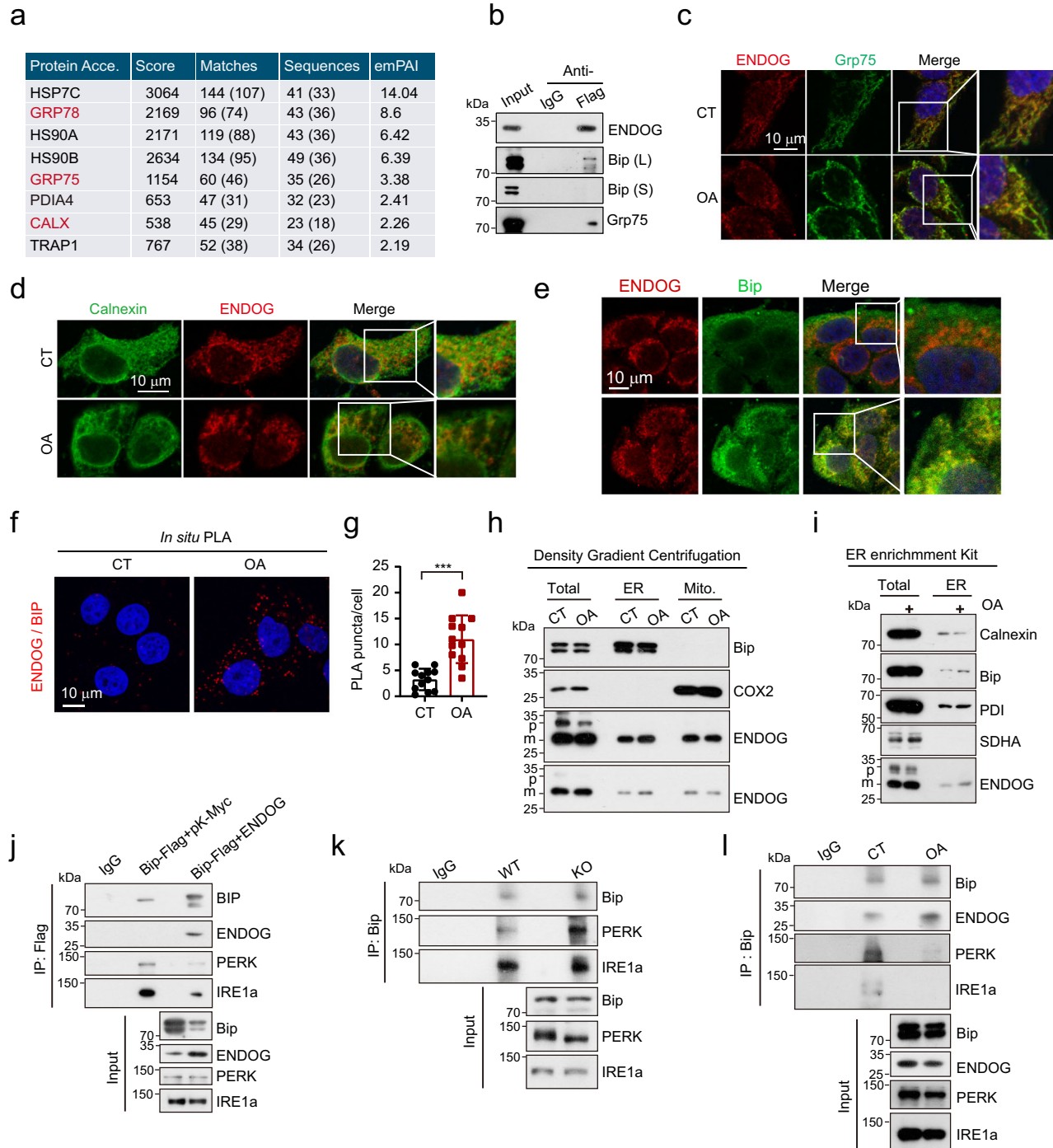

**Fig. 6 | ENDOG activates ER stress by translocating to the ER and binding with Bip. a** ENDOG binding proteins detected by IP-MASS. Protein Acce.: Protein accession. EmPAI: exponentially modified protein abundance index. **b** Co-IP analyses. HepG2 cells were transfected with ENDOG-Flag plasmid for 48 h. L: long exposure; S: short exposure. **c–e** Representative images of costaining of ENDOG with Grp75/ Calnexin/Bip after 200 μM oleic acid treatment for 24 h. **f, g** Representative images of in situ proximity ligation assay (PLA) and quantitative results of PLA puncta per cell. HepG2 Cells were treated with 200 μM oleic acid for 24 h. *n* = 12 biologically independent samples. **h** Subcellular fraction isolation by density gradient centrifugation. HepG2 cells were treated with 200 μM oleic acid for 24 h. ER, endoplasmic reticulum fraction; Mito, mitochondrial fraction; p, precursor ENDOG; m, mature ENDOG. **i** The ER location of ENDOG is analyzed by ER enrichment Kit. HepG2 cells were treated with 200 μM oleic acid for 24 h. Total, total cell lysis; ER, endoplasmic reticulum enriched fraction. **j** Co-IP analyses. HepG2 cells were cotransfected with Bip-Myc and pK-Myc/ENDOG plasmids for 48 h. **k** Endogenous Co-IP analyses in wild-type and ENDOG knockout HepG2 cells. **l** Endogenous Co-IP analyses. HepG2 cells were treated with 200 μM oleic acid for 24 h. Three experiments were repeated independently with similar results in (**b, c, d, e, h, i, j, k, l**). Statistical significance was determined by unpaired Student's t-test (two-tailed) in (**g**); error bars are mean ± SD. Source data and exact *P* value were provided in a Source data file. \*\*\**P* < 0.001.

ENDOG to translocate to the cytoplasm and interact with 14-3-3γ to release TSC2 (an mTORC1 suppressor) and repress the mTORC1, which promoted autophagy[23]. Here, we found that oleic acid treatment also causes the release of ENDOG from mitochondria and binds with 14-3-3γ, which reduces the interaction between 14-3-3γ and Rictor. It has been reported that 14-3-3 negatively regulates mTORC2-AKT by binding with Rictor[27,40]. These data demonstrated that ENDOG promoted mTORC2-AKT-ACLY activation by competitively binding with

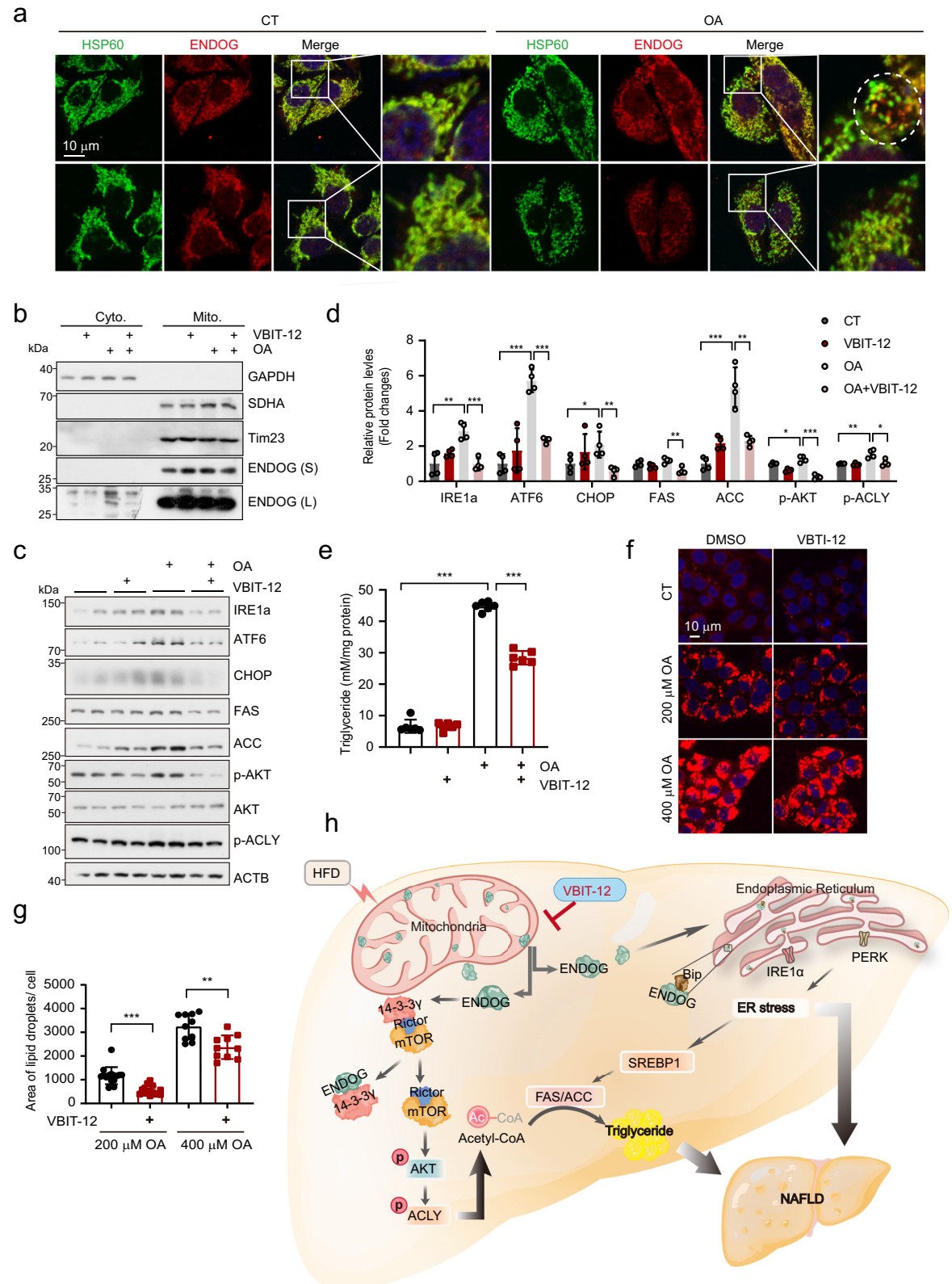

14-3-3γ, which disassociated from Rictor. Besides, We found that loss of ENDOG promoted the activation of mTORC1 and repressed mTORC2 both in the normal and high-fat diet conditions (Fig. S24).

Thus, our previous[23] and the present work found that under stresses (starvation or oleic acid treatment), ENDOG released from mitochondria to the cytoplasm to suppress mTORC1 or active mTORC2, respectively.

It has been reported that silencing ENDOG represses the phosphorylation of AKT (Ser 473) and GSK3β, as well as cell proliferation[41,42]. Consistently, we also found that ENDOG knockout reduced the phosphorylation of AKT, GSK3β and ACLY. AKT-mediated phosphorylation of ACLY is essential for its activation and the production of acetyl-CoA[10]. ENDOG knockout significantly repressed the AKT-ACLY axis and reduced acetyl-CoA production and lipid accumulation in vivo and

**Fig. 7 | Inhibiting the release of ENDOG represses oleic acid-induced lipid accumulation. a, b** Representative images of costaining of HSP60/ENDOG and subcellular fraction isolation. HepG2 cells were pretreated with 20 μM VBIT-12 (VDAC inhibitor) for 4 h and then treated with 200 μM oleic acid for 24 h. The red puncta in the dotted circle indicate that ENDOG is released from the mitochondria. S, short time exposure; L, long time exposure. **c, d** Western blot and quantitative results of ER stress-related proteins, FAS/ACC, and AKT-ACLY activation. HepG2 cells were pretreated with 20 μM VBIT-12 (VDAC inhibitor) for 4 h and then treated with 200 μM oleic acid for 24 h. $n = 4$ biologically independent samples. **e–g** Measurement of triglycerides, Nile red staining, and the quantitative results of lipid area. HepG2 cells were pretreated with 20 μM VBIT-12 (VDAC inhibitor) for 4 h and then treated with 200 μM oleic acid for 24 h. $n = 4$ biologically independent samples for triglycerides measurement and 10 for Nile red staining. **h** Schematic

diagram of ENDOG mediated NAFLD under the HFD chow feeding. HFD promotes the release of ENDOG from the mitochondria. Cytoplasm located ENDOG competitively binds with 14-3-3γ, which makes 14-3-3γ disassociate from Rictor and activates the mTORC2-AKT-ACLY axis, resulting in the production of acetyl-CoA. In addition, cytoplasmic ENDOG also translocates into the ER and interacts with Bip, which releases IRE1a and PERK to activate ER stress and ACC/FAS expression. Increased acetyl-CoA and lipid synthesis enzymes (ACC/FAS) enhanced lipid accumulation. Finally, increased triglyceride and ER stress promote the development of HFD-induced NAFLD. Three experiments were repeated independently with similar results in (**a, b**). Statistical significance was determined by unpaired Student's t-test (two-tailed) in (**d, e, g**); error bars are mean ± SD. Source data and exact *P* value were provided in a Source data file. *$P < 0.05$; **$P < 0.01$; ***$P < 0.001$.

in vitro. AKT phosphorylates GSK3β at Ser9 (an inactivated form of GSK3β) and inhibits the activity of GSK3β. Our previous work[23] demonstrated that GSK3β-mediated phosphorylation of ENDOG increased the binding between ENDOG and 14-3-3γ, which released TSC2 to suppress mTORC1. Indeed, overexpression of GSK3β enhanced ENDOG-induced mTORC1 repression and mTORC2 activation (Fig. S25a, b). Consistent with our previous work, GSK3β-mediated phosphorylation of ENDOG at T128 and S288 enhanced the binding between ENDOG and 14-3-3γ, and then released TSC2 and Rictor, respectively (Fig. S25c, d). TSC2 represses mTORC1 by converting RHEB-GTP (an activator of mTORC1) to the inactive form of RHEB-GDP[43]. Rictor, a subunit of mTORC2, has been reported to reduce mTORC2 activity when binding with 14-3-3[27]. Besides, it has shown that the TSC1-TSC2 complex is required to properly activate mTORC2 independent of its GTPase-activating protein activity toward Rheb[44,45]. Thus, the binding between ENDOG and 14-3-3γ, which releases TSC2 and Rictor, could suppress mTORC1 and mTORC2. AKT-mediated phosphorylation of GSK3β inhibited glycogen synthesis[46]. AKT also regulates cell glycolysis by phosphorylating glycolysis enzymes, such as GLUT1, HK2, and PFKFB[47]. However, in our model, we found that loss of ENDOG did not affect cell glycolysis, as indicated by the ECAR and glycolysis enzyme expression (Fig. S26).

AKT phosphorylates various downstream substrates, including ACLY, mTOR, p70S6K, and GSK3β. AKT repressed autophagy by activating mTORC1. AKT promoted lipid accumulation in wild-type, ENDOG overexpressed, and knockout cells (Fig. S27a, d), possibly due to AKT-mediated ACLY activation. However, in wild-type, ENDOG overexpressed or knockout cells, we found overexpression of AKT had little influence on the activation of GSK3β and mTORC1 but dramatically increased the phosphorylation of p70S6K (Fig. S27b, c and e, f) and increased phosphorylation of ACLY in ENDOG knockout cells (Fig. 2o). Thus, our model considered that ENDOG-mediated activation of mTORC2-AKT mainly promoted ACLY activation and lipid synthesis but had little influence on GSK3β phosphorylation and mTORC1 activation. Our results demonstrated that GSK3β-mediated phosphorylation of ENDOG enhanced its interaction with 14-3-3γ, which weakened the binding between 14-3-3γ with TSC2 and Rictor. The released TSC2 and Rictor repressed mTORC1 or activated mTORC2, respectively. The activation of AKT in ENDOG overexpressed cells promoted the phosphorylation of GSK3β, which repressed the kinase activity of GSK3β, acting as a negative feedback loop of ENDOG function.

Our previous data demonstrate that ENDOG promoted autophagy. Autophagy is a negative lipid accumulation regulator by lipophagy, degrading lipid droplets by autolysosomes. Lipophagy was first observed in hepatocytes by co-staining of autophagic proteins LC3B and LAMP1 with lipid droplets[48]. We found little co-localization of lipid droplets with LC3B and LAMP1 in wild-type, ENDOG overexpressed, and knockout cells (Fig. S28a, b), indicating that lipophagy may not be the primary manner of lipid degradation in our present model. ATGL and HSL, two rate-determining enzymes of lipolysis, had increased in ENDOG overexpressed cells and reduced in ENDOG knockout cells

(Fig. S28c–f). It is possible increased ATGL/HSL may be caused by the negative feedback of ENDOG-mediated lipid accumulation. Our previous work demonstrated that ENDOG-promoted autophagy also via its endonuclease activity-mediated DNA damage response. However, we found that overexpression of ENDOG-EM (endonuclease activity null mutant) promoted AKT-ACLY activation, ER stress and lipid accumulation, similar with ENDOG overexpression (Fig. S29a, b, and d–e). Moreover, the endonuclease activity mutation of ENDOG did not affect the binding of ENDOG with 14-3-3γ and Bip (Fig. S29c, f). Furthermore, the endonuclease activity inhibitors of ENDOG (PNR-3-80 and PNR-3-82) had no effects on lipid accumulation after oleic acid treatment (Fig. S29i, j). These data indicated that ENDOG promotes AKT-ACLY activation, ER stress and lipid accumulation through its endonuclease activity independent manner.

In the present study, male and female mice were fed HFD. However, we found that loss of ENDOG alleviated HFD-induced NAFLD in female but not in male mice (Figs. 3 and S9, 10, 12). By the GTT and ITT assays, we found that ENDOG only regulates the insulin sensitivity of HFD-fed female mice (Fig. S12), which may be one reason for the difference in phenotype of female and male mice. Besides, ENDOG may also regulate the hormone level or downstream signaling, which needs further study.

Almost 99% of mitochondrial proteins are encoded by nuclear genes, synthesized as precursors on cytosolic ribosomes and then translocated into mitochondria by passing through the translocase on the outer mitochondrial membrane[49,50]. The nuclear gene encodes ENDOG as a precursor with 297 amino acids. Amino acids 1 to 48 at the N-terminal of ENDOG consist of the mitochondrial targeting sequence and are removed to form mature ENDOG when imported into mitochondria[12]. In this study, we found that the mature form of ENDOG was mainly located in the mitochondria, but we also detected it in the ER fraction or enrichment region, indicating that ENDOG may translocate to the ER after its release from mitochondria. AIF, another mitochondrial intermembrane space protein that is also released from mitochondria and further translocated to the nucleus to trigger DNA fragmentation and initiation of apoptosis[51,52], was detected in fractions of the endoplasmic reticulum, mitochondria-associated membranes (MAM) and mitochondria[53]. Further study showed that AIF was imported from the ER to the mitochondria via MAM and transport vesicles[53]. However, we detected both precursors or mature forms of ENDOG in total, but only the mature form of ENDOG was detected in the ER fraction in the control and oleic acid treatment groups, suggesting that ENDOG, unlike AIF, was imported into mitochondria, and then translocated to the ER when it was released during stress stimulation.

Previous studies have shown that proapoptotic proteins (e.g., Bax, Bid, and Bak) bind to the outer mitochondrial membrane, leading to outer-membrane permeabilization and mitochondrial permeability transition pore (MPTP) opening. This is followed by the release of cytochrome c, ENDOG, and AIF from the mitochondrial intermembrane space to the cytosol and translocation to the nucleus, leading to apoptotic cell death[54,55]. Z-IETD-FMK (caspase-8 inhibitor)

represses the activation of Bid and can block the release of ENDOG and AIF[23,39,56]. Mitochondrial Ca$^{2+}$ accumulation, adenine nucleotide depletion, increased phosphate concentration or ETC-generated ROS promote MPTP opening. MPTP consists of a voltage-dependent anion channel (VDAC), adenine nucleotide translocator (ANT), and cyclophilin D (CyPD)[57–59]. Thus, blocking the formation of the MPTP could repress the MPTP-dependent release of ENDOG and nuclear translocation[19]. Blocking VDAC with a VDAC antibody could abolish the mitochondrial localization of BNIP3 and prevent ENDOG release[36]. Consistently, we found that treatment with the VDAC oligomerization inhibitor VBIT-12 could block the release of ENDOG, followed by suppression of oleic acid-induced activation of ER stress, expression of FAS, ACC, p-AKT, p-ACLY, and lipid accumulation.

Bip is an ER resident protein that binds with ER stress sensors under normal conditions. Increased misfolding/unfolding proteins competitively bind with Bip and lead to the release of Bip from ER stress sensors and activate ER stress[34]. ER stress regulates lipid metabolism through multiple pathways. The IRE1a-XBP1 axis regulates gene expression related to lipid biosynthesis and triglyceride contents, including ACC, FAS, and SCD1[31,60,61]. Furthermore, XBP1 upregulates the transcription of fatty acid synthase genes by binding to the promoter of SREBP1[60,62]. PERK activation promoted the phosphorylation of eIF2a, reduced the translation of Insig1, and promoted SREBP1 maturation and nuclear localization to initiate the transcription of lipid synthesis genes[63]. ATF6 promoted the expression of ACC and FAS in MEFs. In addition, ATF6 also regulates lipid metabolism by promoting the expression of fatty acid oxidation genes (*CPT1*, *CPT2*, *ACOX1*, and *PPARA*) and VDVL formation genes (*MTTP*, *PDI*, and *APOB*)[64]. In our study, ENDOG deficiency reduced the expression of IRE1a, PERK, and ATF6, as well as the splicing of XBP1. In addition, we found that ENDOG knockout decreased the expression of SREBP1, ACC, and FAS but not the expression of fatty acid uptake genes (*CD36*, *FATP2*, and *FATP5*) and β-oxidation genes (*CPT1*, *CPT2*, and *ACOX1*). Taken together, our data demonstrated that on one hand, ENDOG deficiency reduced the lipid synthesis substrate acetyl-CoA through the mTORC2-AKT-ACLY axis; on the other hand, loss of ENDOG repressed the expression of lipid synthesis enzymes via translocation to the ER and subsequent activation of ER stress. The reduced lipid accumulation and ER stress in ENDOG knockout mice livers eventually alleviated HFD-induced NAFLD in females.

## Methods

### Mice and *C. elegans*
All mice were handled according to the China Animal Welfare Legislation guidelines and as approved by the Animal Ethics Committee of Jinan University. ENDOG-knockout mice (C57BL/6) and *Endog* $^{flox/flox}$ mice were generated by Cyagen Biosciences (Guangzhou) Inc. using the CRISPR/Cas9 system. To generate a liver-specific deletion of ENDOG, *Endog* $^{flox/flox}$ mice were crossed with albumin-Cre transgenic mice. Mice were maintained on a 12 light/12 dark cycle (light on 7 AM to 7 PM), at a room temperature of 22 °C ± 2 °C, humidity of 50% ± 5%, with free access to food and water. For HFD treatment, male and female littermate ENDOG systemic knockout (*Endog*$^{+/-}$ and *Endog*$^{-/-}$) and liver-specific knockout (*Endog*$^{flox}$ and *Endo*$^{LKO}$) mice were fed a HFD (D12492, Research Diets) for 16 weeks at the age of 6 weeks. For liver lipid quantifications in Fig. 1, male littermate ENDOG systemic knockout (*Endog*$^{+/-}$ and *Endog*$^{-/-}$) were starved for 24 h. For TM treatment, 8 ~ 10-week-old female mice were fasted for 24 h and then intraperitoneally injected with tunicamycin (2 mg/kg body weight) for 24 h.

The wild-type N2 (Bristol) and ENDOG mutant cps-6 (tm3222) *C. elegans* strains were obtained from the Caenorhabditis Genetic Center (CGC) (University of Minnesota, USA) and the National Bioresource Project (NBRP), respectively. For high-fat diet analyses, the N2 and ENDOG mutant *cps-6 (tm3222)* L1 phase nematode strains were treated with a high-fat diet (the cooked random egg yolk was ground and mixed with OP50 (volume ratio, 1:5)) until L4 larvae and then for oli o staining.

### Cell culture
Hepatocytes (HepG2 (SCSP-510), MHCC97H (SCSP-5092), HCC-LM3 (SCSP-5093), Huh7 (SCSP-526)) and 293 T (SCSP-502) were obtained from the Shanghai Cell Bank, Type Culture Collection Committee, Chinese Academy of Sciences. ENDOG knockout cell lines were generated using the CRISPR/Cas9 system with two sgRNAs (sgRNA1: CCGGGCGAGCTGGCCAAGTA and sgRNA2: CGACTTCCGCGAGGAC GACT). Cells were transfected with pRX-001-sgRNA-ENDOG, and positive clones were selected with 1μg/ml puromycin, PCR genotyping and sequencing were used to choose ENDOG knockout clones. Hepatocytes and 293 T cells were maintained in DMEM (Gibico, C11995500BT) supplemented with 10% fetal bovine serum (ExCell Bio, FSD500), 1% penicillin and streptomycin (Gibico,15140-122) and incubated at 37 °C in a 5% CO$_2$ incubator.

### Western blotting and immunoprecipitation
Cells or 20 mg liver tissue were lysed with RIPA buffer (supplemented with PMSF and phosphatase inhibitor cocktail). Protein samples (20 μg) were loaded for western blotting. For immunoprecipitation, two 10-cm dish cells were lysed with Western/IP buffer (supplemented with PMSF and phosphatase inhibitor cocktail). Immunoprecipitation was performed as previously described[23]. Precipitates were boiled and loaded onto SDS-PAGE gels for western blot analyses. For ENDOG western blotting, the PVDF membrane was incubated with the primary antibody of ENDOG (CST, Cat: #4969, 1: 1000 diluted) overnight at 4 °C and then incubated with the HRP-Rabbit secondary antibody (Jackson Immunoresearch, Cat: 111-035-003, 1:5000 diluted) at room temperature for 2 h. Other primary antibody information are presented in the Supplementary Information in Table S1.

### Nile red, BODIPY, and Oil Red O staining
After the treatments, cells were fixed with 4% paraformaldehyde at room temperature for 15 min. For Nile red and BODIPY staining, cells were washed with PBS after fixation and stained with 1 μg/ml Nile red or 0.25 μg/ml BODIPY for 15 min. For liver-frozen sections, sections were air-dried for 30 min at room temperature and then fixed with cold acetone for 15 min. After washing with PBS, sections were stained with 0.25 μg/ml BODIPY for 15 min. After costaining with DAPI, cells or sections were imaged with a Leica TCS SP8 confocal microscope. For oil red O staining, cells were washed extensively with ddH$_2$O after fixation and then stained with oil red O working solution at room temperature for 30 min. Images were acquired by a Zeiss microscope. The lipid droplet area was measured with ImagePlus 6.0 software.

### In situ proximity ligation assay (PLA)
A proximity ligation assay was performed using the Duolink In situ Kit (Table S2) according to the manufacturer's instructions. Briefly, after treatment with 200 μM oleic acid for 24 h, cells were fixed with 4% PFA for 20 min, permeabilized with 0.2% Triton X-100 for 10 min and blocked with Duolink blocking solution for 60 min at 37 °C. Then the cells were incubated with anti-ENDOG and anti-Bip antibodies diluted in Duolink antibody diluent for 2 h at room temperature, washed twice with wash buffer A, and incubated with PLA probes (PLUS and MINUS) for 60 min at 37 °C. A ligation reaction was performed to hybrid connector oligos join the PLA probes for 30 min at 37 °C, and PLA signal amplification was performed for 100 min at 37 °C. Finally, the cells were washed with buffer B and stained with DAPI. Images were taken with a Leica TCS SP8 confocal microscope. Positive PLA puncta were analyzed by ImagePlus 6.0.

### Double immunofluorescence staining
Cells were fixed with 4% formaldehyde for 15 min and permeabilized with 0.2% Triton X-100 for 10 min at room temperature. Then, the cells were blocked with PBST containing 2% BSA and 10% goat serum for 1 h

at room temperature. For two different host antibodies, two primary antibodies were mixed and incubated with cells for 2 h at room temperature. After extensive washing with PBST, the cells were incubated with a mixture of Alexa Fluor® 594-AffiniPure goat antirabbit and Alexa Fluor® 488-AffiniPure goat anti-mouse for 1 h at room temperature. After washing with PBST, the cells were stained with DAPI and imaged. ImageJ software was used to analyze the colocalization of two different proteins. For ENDOG immunofluorescence staining, cells were incubated with the primary antibody of ENDOG (NOVUS, Cat: IMG-5565-2, 1: 100 diluted) overnight at 4 °C and then incubated with the Alexa Fluor® 488-AffiniPure Goat Anti-Rabbit (Jackson Immunoresearch, Cat: 111-545-144) at room temperature for 2 h. After extensive washing, cells were stained with DAPI and imaged.

## Subcellular fractionation isolation

Subcellular fractionation isolation was performed as previously described[23]. Cells were digested after treatments and centrifuged at 400 g at 4 °C for 5 min. The cell pellet was permeabilized with 10 mg/ml digitonin in PBS at room temperature for 10 min and centrifuged at 10,000 g at 4 °C for 10 min. The supernatants were collected as the cytosolic fraction. The pellets were resuspended in mitochondrial lysate buffer[23] on ice for 30 min and then centrifuged at 10,000 g at 4 °C for 20 min. The supernatants which contain mitochondrial fractions were collected.

## ER fraction isolation and ER enrichment

ER fraction isolation was performed as previously described[65]. After 200 µM oleic acid treatment for 24 h, cell fractionations were isolated and analyzed by SDS-PAGE. The ER isolation assay was performed as described in a previous study[66]. Briefly, after treatment, cells were harvested by trypsin digestion and suspended in 1 × MTE buffer (270 mM D-mannitol, 10 mM Tris-base, 0.1 mM EDTA, pH 7.4). Cells were sonicated 5 s on/5 s off three times on ice. The lysed cells were centrifuged at 1400 g for 10 min at 4 °C and 100 µl of supernatant labeled "total protein" was collected. The remaining supernatant was transferred into a new tube and centrifuged at 15,000 g for 10 min at 4 °C. After centrifugation, there should be a yellow-brown pellet containing crude mitochondrial protein at the bottom of the tube. The supernatant was transferred and layered onto the ER sucrose gradient (2 ml 2.0 M, 3 ml 1.5 M, 3 ml 1.3 M), 1 × MTE buffer was added to the top of the ER gradient. The ER gradient was ultracentrifuged at 152,000 g for 70 min at 4 °C and a 0.5 ml volume of the large band at the interface of the 1.3 M sucrose gradient layer was collected into a new Beckman polyallomer tube for another ultracentrifugation at 126,000 g for 45 min. The supernatant was discarded, the pellet was dried, and then the pellet was resuspended in PBS and labeled "ER". The yellow-brown crude mitochondrial protein was carefully washed with 1 × MTE buffer three times, and then the mitochondrial supernatant was transferred and layered onto the mitochondrial sucrose gradient (1 ml 1.7 M, 1.6 ml 1.0 M), 1 × MTE buffer was added to the top of the mitochondrial gradient. The mitochondrial gradient was ultracentrifuged at 40,000 g for 22 min at 4 °C, and 0.4 ml of the large band at the interface of the 1.7 M and 1.0 M sucrose layer was collected into a new tube. The sample was resuspended in 1 × MTE buffer, and then centrifuged at 15,000 g for 10 min at 4 °C. The supernatant was discarded, the pellet was dried, and then the pellet was resuspended in PBS and labeled as "mitochondria".

ER enrichment was performed by the ER enrichment Kit (Invent Biotecnologies, Inc., Cat: NO. ER-036). Briefly, after 200 µM oleic acid treatment, $3 \times 10^6$ cells were collected by centrifugation at 550 g for 5 min. Cell pellets were frozen at −80 °C for 10 min and then resuspended in 550 µl buffer A, and the tube was vortexed vigorously for 30 s. The suspension was transferred to a filter cartridge and centrifuged at 16,000 g for 1 min. Then, the pellets were resuspended and centrifuged at 2000 g for 5 min. All supernatants were collected and

centrifuged at 4 °C, 8000 g for 10 min. After centrifugation, 400 µl of supernatant was transferred to a fresh tube, and 40 µl of buffer B was added. After gentle vortexing, the mixture was incubated at 4 °C for 45 min and centrifuged at 8000 g for 10 min. The pellets were suspended in 400 µl of buffer A and vigorously vortexed for 30 seconds. Then 40 µl buffer C was added and mixed well. After incubation at room temperature for 15 min, the tube was centrifuged at 8000 g for 5 min. Subsequently, 400 µl of supernatant and 400 µl of buffer D were added to a fresh tube and incubated at 4 °C for 20 min. After incubation, the mixture was centrifuged at 10,000 g for 10 min. The pellet was "ER enrichment". Then 100 µl PBS was then added to the pellet and stored at −80 °C.

## qRT-PCR

RNA was isolated using RNAiso Plus reagent following the manufacturer's protocol. cDNA synthesis was performed using the AB Script II cDNA First Strand Synthesis Kit. SYBR Green Select Master Mix was used for qPCR on a CFX96 real-time system, and data were analyzed by means of CFX Manager Software using the $2^{-\Delta\Delta CT}$ method, with Actb used as the internal control. Primer sequences for the target genes are provided in the Supplementary Information in Table S3.

## ACLY and ENDOG knockdown

ACLY and ENDOG shRNA (sequences in Tables S4 and S5) were inserted into the pLKO.1 vector, which was then transfected into 293 T cells. After 48 h of transfection, the virus was collected and used to infect hepatocytes cells. Stable ACLY or ENDOG knockdown cell lines were selected with 1 µg/mL puromycin.

## H&E staining and hepatic steatosis analyses

Paraffin-embedded liver sections were stained with hematoxylin-eosin (H&E) according to the manufacturer's instructions. For steatosis analyses, 4 different scores were used: 0, steatosis <5%; 1, 5% < steatosis <33%; 2, 33% < steatosis <66%; and 3, steatosis >66%[67].

## Extracellular acidification rate (ECAR) analyses

Cells (10,000) were plated in each XF96-well microplate well (Seahorse Bioscience) followed by ECAR measurement at 37 °C using an XF96 Analyzer (Seahorse Bioscience). Then, final concentrations of 10 mM glucose, 1.5 µM oligomycin and 50 mM 2-DG from the Seahorse XF Cell Glycolysis Stress Test kit (Agilent, 103020-100) were added to each well. XF data were normalized to the protein concentration of each well, which was detected by BCA analyses.

## GTT and ITT assay

After HFD chow, male and female mice were performed with GTT and ITT assay. For the GTT assay, mice were fasted overnight for 16 h with free access to water. Each mouse was weighed, and fasting blood glucose was measured by tail blood with a One Touch Ultra Glucometer (Lifescan Canada Ltd.) the next morning. 1.5 g/kg body weight of glucose was intraperitoneally injected into each mouse, and then blood glucose was measured at 15, 30, 60, 90, and 120 min after glucose intraperitoneally injection. For the ITT assay, mice were fasting for 6 h. Insulin solution (0.75 U/kg bodyweight) was administered intraperitoneally. Blood glucose was measured 15, 30, 60, 90, and 120 min after insulin injection. The data were plotted as blood glucose concentration (mM) over time (minutes).

## Statistical Analyses

All statistical analyses were performed using GraphPad Prism software. Experiments were performed at least 3 times independently under similar conditions. All statistical results are presented as the mean ± standard deviation (S.D.). The statistical significance between two groups was calculated using the two-tailed, unpaired Student's t test; defined as $*p < 0.05$, $**p < 0.01$, and $***p < 0.001$.

## Data availability

All data supporting the findings of this study are available within main text, supplementary information and Source data. The RNA-seq data generated in this study had been deposited in the Sequence Read Archive (SRA) database under BioProject accession codes PRJNA987814 and PRJNA987815. The western blot gels and graph data are included as Source Data. Source data are provided with this paper.

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

## Acknowledgements

This work was supported by the National Key R&D Program of China (2021YFA0804903 to Q.Z.), the National Natural Science Foundation of China (Grants. 32270810 to WW and 32170772 to JT), the 111 Project (B16021 to Q.Z.), the Natural Science Foundation of Guangdong Province (Grants. 2021A1515011227 to W.W.), and the Singapore Ministry of Health's National Medical Research Council (OFIRG22jul-0007 to H.T.). Q.Z. also gratefully acknowledges the support of K.C. Wong Education Foundation.

## Author contributions

Q.Z. and W.W. designed experiments. W.W. and J.T. performed the experiments and data analyses. X.-m L., M.L., X.-j L., W.G., Y.L., and W.D. performed experiments. L.H., W.L., and H.T., critically read the paper. Q.L. and Y.W. provided technical support and RNA-Seq data analyses. Q.Z. and W.W. wrote the paper.

## Competing interests

The authors declare no competing interests.
