## [Peer Review File · Nature Communications]

REVIEWER COMMENTS

Reviewer #1 (Remarks to the Author):

The paper titled "Cytoplasmic translocated Endonuclease G promotes NAFLD by enhancing the mTORC2-AKT-ACLY axis and ER stress" by Wang et al. presents interesting findings on EndoG's role in NAFLD. The authors demonstrate that EndoG translocates to the ER, interacts with BiP, and activates ER stress through PERK and IRE1a. Moreover, EndoG interacts with 14-3-3g to activate the MTORC2-AKT pathway, resulting in lipogenic program activation and NAFLD.

However, the authors' previous work (Nat Commun 2021, 12:476) has demonstrated conflicting events in similar hepatic models. Specifically, in an earlier study, the authors showed that EndoG promotes autophagy by inhibiting MTORC1 via its interaction with 14-3-3g. These events were dependent on GSK3b-mediated phosphorylation of EndoG, and there were strikingly fewer lipid droplets. In the current study, the authors demonstrate that EndoG promotes NAFLD by activating the lipogenic pathway via ER stress and MTORC2-AKT pathways. The following points need clarification for a better understanding of EndoG regulation of NAFLD:

1. It is essential to study GSK3b-MTORC1-autophagy in normal chow-fed and HFD-fed WT and EndoG KO and LKO mice as both models are similar.
2. It is crucial to study GSK3b-EndoG (phosphorylation)-MTORC1/MTORC2-AKT and AKT-GS3b regulation in these mice livers as GSK3b is a bona fide target of AKT.
3. It is also important to study as autophagy in general is associated with decreased lipid accumulation by lipophagy. However, in this case lipogenic pathway can lead to more lipid formation. Additionally, what happened to other lipolytic pathways such as lipolysis by classical lipases when beta-oxidation is not changed (Figure 5A) and autophagy is inhibited (as per the old study)?
4. Although EndoG KOs showed less lipid accumulation, what happened to hepatic inflammation in HFD mice when autophagy is also inhibited (as per the old study)? Autophagy has been shown to be key in regulating inflammation during NAFLD.
5. There are other transcription factors regulated by ER stress sensors such as PPARg, ChREBP, SREBP1c, SREBP2, FABP4, CRTC2, CREBH, etc. what happened to these or SREBP1 is the only one affected by Endo G-associated ER Stress? Also, SREBP1c is the predominant isoform in the liver regulating hepatic lipogenesis so including SREBP1c along with the others is more appropriate.
6. FASN only partially regulates oleic acid-induced TG synthesis (lipid droplets) and also requires SCD1 activity as oleic acid is a mono-unsaturated fatty acid and not fully dependent on FASN activity (majorly regulating saturate fat palmitic acid formation). Please also include SCD1 and DGATs expression to show if the complete pathway is upregulated.
7. The authors also showed that insulin treatment restored lipid accumulation, and EndoG activated AKT in the liver. How are the blood glucose and insulin levels in the mice at basal and under starvation? it is important to analyze fasting blood glucose, glucose tolerance, and Insulin tolerance in normal chow-fed and HFD-fed WT and KO/LKO mice. This is especially important to understand if targeting EndoG leads to an improved fatty liver but interferes with other important pathways such as autophagy, insulin resistance, hepatic inflammation etc.
8. The authors need to comprehend their old findings with the new (as both models are the same) once in a way to better understand and link them together.
9. Lastly, any specific reason for choosing female mice for the NAFLD study as male mice are the preferred choice for such metabolic studies to reduce the hormonal influence? What happens in the male EndoG mice on HFD?

The Reviewer believes that the comments will be helpful to increase clarity and improve the undersatding of current work.

Reviewer #2 (Remarks to the Author):

In this manuscript, Wang et al. described their observation that ENDOG translocates to the ER, binds with Bip resulting in release of IRE1a/PERK that activates the ER stress response, promoting the expression of SREBP1, ACC, and FAS. Using in vitro and in vivo approaches they have beautifully shown that loss of ENDOG suppresses acetyl-CoA production and lipid synthesis, along with reducing ER stress, which eventually alleviates HFD-induced NAFLD. Although the work of the manuscript is very attractive there are some major concerns.

Major Comments

1. Previous study from Wang et al, has described, "Endonuclease G promotes autophagy by suppressing mTOR signalling and activating the DNA damage response, <https://doi.org/10.1038/s41467-020-20780-2> ."which the reviewer felt a bit contradicting, how do the authors justify two opposite mechanism in once cell type, there need a strong explanation to it.
2. The authors should evenly show the densitometry as some figures have it and some don't, which is a bit of a concern.
3. the authors should clearly state the cell lines that has been used and have they validated that cell line or have they checked the reproducibility in any other cell line.
4. Some of the blots looked overexposed, the authors should take a careful look at the blots, supplementary 11 is just an example.
5. The reviewer kept wondering as authors have not clearly mentioned regarding the improvement in the glucose tolerance or insulin levels, it would be good to show that data.

Reviewer #3 (Remarks to the Author):

The work by Wang, Tan and colleagues investigates the role of Endonuclease G (ENDOG) in liver physiology and its contribution to the development of non-alcoholic fat liver disease (NAFLD). Results illuminate a two-pronged mechanism that together sustains the accumulation of intracellular lipid storage by ENDOG. In fact, ENDOG promotes the de novo production of cytoplasmic acetyl-CoA, a building block for the synthesis of lipids, and induces ER stress to activate SREBP, a meter regulator of lipid biosynthesis.

The study builds from prior work on ENDOG and stems from side observations on Endog-ablated cells. The study significantly adds to our knowledge of ENDOG function and is of significant value for multiple research fields being at the intersection between liver pathophysiology, cellular metabolism, signaling.

The work is well executed and deploys a number of biochemical and cell biology techniques, as well as multiple Endog-deficient animal models. Experiments are non redundant and results extremely sharp and clear.

Ultimately, the data supports authors' conclusions, although proposed mechanisms do have gray areas that could be addressed.

I believe that the robustness of the work might improve if the authors addressed these concerns:

- 1) the observation that ENDOG-KO reduces LDs is a cornerstone of the manuscript. Given its key position, a better characterization would be appreciated. Can the authors stain for LD markers? Is PLIN2 expression also reduced? Can the authors rule out those bodies are in fact lipid-laden endosomes?
- 2) The manuscript posits that ENDOG-KO limits lipid synthesis and deposition onto LDs. However, the data do not actually rule out the possibility that LDs are smaller because lipids are mobilized and oxidized more. Even if acetyl-CoA levels are augmented, we do not know where that comes from and the directionality of metabolic fluxes. And the result that ACLY depletion decreases LDs upon OA addition might actually indicate that enhanced fatty acid oxidation is in play. I recommend that the authors test the levels of FAO in ENDOG-KO cells and it would be appreciated if they could perform carbon tracing to determine the source of acetyl-CoA (I acknowledge this might be challenging and beyond the scope of revision)
- 3) The relationship between ENDOG and ER stress is interesting and significantly adds to existing literature. However, the mechanism could be better characterized. The association of ENDOG with Bip

and Gpr75 seems weak and is minimally characterized biochemically and the possibility that ER stress is a secondary consequence of LD decrease has been neglected (e.g.: LDs sequester toxic lipids and alleviate ER stress; PMID: 19081072; 25829424). The manuscript would improve if this possibility is discussed and tested.

I also have a couple of conceptual concerns on which I would appreciate a comment from authors.

1) acetyl-CoA levels are down in ENDOG-KO cells and mouse livers (Fig 2C-D), and this is ascribed to reduced flux through ACLY. However, a large fraction of the acetyl-CoA pools in hepatocytes is produced from acetate. Have the authors tested the possibility that ENDOG regulates acetate metabolism? or acetate recycling (from de-acetylation reactions)?

2) Cells were supplemented with exogenous acetyl-CoA and rescue in LD and TG content was observed (Fig 2H-J). This puzzles me because acetyl-CoA does not cross the plasma membrane and no transporter is known to my knowledge. In fact, this is the very first time I see someone attempt a rescue experiment with exogenous supplementation of acetyl-CoA. Can the authors comment on that?

Reviewer #4 (Remarks to the Author):

Zhou and co-workers investigated whether EndoG, a mitochondrial protein, has a role in the cytosol. They found that loss of EndoG decreased acetyl-CoA production, lipid synthesis, and ER stress which eventually relieved HFD-induced NAFLD. This is a high-quality study that provides insights into the HFD-induced NAFLD. Although the observed effects of cytoplasmic EndoG are significant, it remains unclear whether they are influenced or not by the endonuclease activity of the protein. It would be important to determine whether loss of the nuclease activity of EndoG affects acetyl-CoA production, lipid synthesis, and ER stress.

Comments:

Fig. 2A-B:

The effects of EndoG on AKT phosphorylations appear to be quite modest. It would be important to quantify the relative intensities of the bands observed in the Western blots against AKT S473 and AKT T308.

Fig. 5E:

Treatment of WT cells with TM appears to increase the PERK signal only weakly. Could you quantify the intensities of the bands observed in the Western blots against PERK in the experiments involving the TM and TG-treated cells?

Point-by-point response

We appreciate the Editor and Reviewers for considering the strengths of our work and for their valuable advice and suggestions for improving this manuscript. We have tried our best to address these points by conducting new experiments and revising the manuscript. Below are our point-by-point responses (blue font) to the reviewers' comments.

REVIEWER COMMENTS

Reviewer #1 (Remarks to the Author):

The paper titled "Cytoplasmic translocated Endonuclease G promotes NAFLD by enhancing the mTORC2-AKT-ACLY axis and ER stress" by Wang et al. presents interesting findings on EndoG's role in NAFLD. The authors demonstrate that EndoG translocates to the ER, interacts with BiP, and activates ER stress through PERK and IRE1a. Moreover, EndoG interacts with 14-3-3g to activate the MTORC2-AKT pathway, resulting in lipogenic program activation and NAFLD.

However, the authors' previous work (Nat Commun 2021, 12:476) has demonstrated conflicting events in similar hepatic models. Specifically, in an earlier study, the authors showed that EndoG promotes autophagy by inhibiting MTORC1 via its interaction with 14-3-3g. These events were dependent on GSK3b-mediated phosphorylation of EndoG, and there were strikingly fewer lipid droplets. In the current study, the authors demonstrate that EndoG promotes NAFLD by activating the lipogenic pathway via ER stress and MTORC2-AKT pathways. The following points need clarification for a better understanding of EndoG regulation of NAFLD:

1. It is essential to study GSK3b-MTORC1-autophagy in normal chow-fed and HFD-fed WT and EndoG KO and LKO mice as both models are similar.

Response: As requested, we detected the phosphorylation of GSK3 β and the substrates of mTORC1 in ENDOG KO and LKO mice under both normal and HFD chow conditions. Moreover, we discussed this part thoroughly in the discussion section of the revised manuscript as follows.

Here, in the ENDOG-KO and LKO mice livers, we found that loss of ENDOG activates mTORC1 signaling in both normal and HFD-fed mice, as shown by the increased

phosphorylation of ULK1, p70S6K and 4EBP1 (Figures S24a-b and e-f). Besides, we detected a decrease of LC3B-II in ENDOG knockout mice livers, indicating that loss of ENDOG repressed autophagy in both normal and HFD-fed mice.

Additionally, we found that ENDOG loss repressed the phosphorylation of AKT at serine 473 in both normal and HFD-fed mice. As ACLY and GSK3 β were substrates of AKT, here we found that both the phosphorylation of ACLY and GSK3 β were repressed in ENDOG-KO and LKO mice livers under normal and HFD chow conditions (Figure S24). These data suggested that loss of ENDOG activated mTORC1 activity and then repressed autophagy in normal and HFD-fed mice, consistent with our previous work.

Meanwhile, loss of ENDOG repressed mTORC2-AKT activation and then suppressed ACLY phosphorylation.

Figure S24

2. It is crucial to study GSK3 β -EndoG (phosphorylation)-MTORC1/MTORC2-AKT and AKT-GS3b regulation in these mice livers as GSK3 β is a bona fide target of AKT.

Response: As suggested, we detected the level of GSK3 β and its phosphorylation in the normal and HFD-fed mice livers and found that ENDOG knockout decreased the phosphorylation of GSK3 β , AKT, and ACYL (Figure S24). To investigate whether GSK3 β -

mediated phosphorylation of ENDOG also affects its role in regulating mTORC1 and mTORC2 pathways, we reintroduced GSK3 β in ENDOG-overexpressed and the control cells. We found that ENDOG repressed the phosphorylation of mTOR, ULK1, p70S6K, and 4EBP1, and overexpression of GSK3 β enhanced the repression role of ENDOG on mTORC1 signaling (Figure S25 a-b). Furthermore, GSK3 β overexpression also increased the phosphorylation level of AKT and ACLY in ENDOG overexpressed cells (Figure S25 a-b), indicating that GSK3 β promoted ENDOG-mediated mTORC1 suppression and mTORC2 activation.

Additionally, we reproduced our previous data and found that overexpression of GSK3 β promoted ENDOG binds to 14-3-3 γ , which reduced the interaction between 14-3-3 γ with TSC2 and 14-3-3 γ with Rictor. However, the enzyme activity deficient GSK3 β (GSK3 β -Mut) could not affect the binding between ENDOG /Rictor with 14-3-3 γ (Figure S25c). While, ENDOG-AA (T128S288 to A128A288), which GSK3 β could not phosphorylate, did not affect the interaction between ENDOG with 14-3-3 γ , and 14-3-3 γ with TSC2 and Rictor (Figure S25d). These data suggested that GSK3 β -mediated phosphorylation of ENDOG enhanced its interaction with 14-3-3 γ , which weakened the binding between 14-3-3 γ with TSC2 and Rictor. The released TSC2 and Rictor repressed mTORC1 or activated mTORC2, respectively. Besides, it has shown that the TSC1-TSC2 complex is required for proper activation of mTORC2 in a manner independent of its GTPase-activating protein activity toward Rheb (Jingxiang Huang, et al, 2008, Mol Cell Biol, PMID: 18411301). Thus, the released TSC2 not only inhibits mTORC1 but also activates mTORC2. Collectively, ENDOG promoted the release of TSC2 and Rictor from 14-3-3 γ , which both activated the mTORC2 signaling.

Figure S25

To investigate the role of AKT-GSK3 β signaling in ENDOG-mediated regulation in the mTORC1 pathway, we overexpressed an activated form of AKT in ENDOG overexpressed and knockout HepG2 cells. We found that overexpression of AKT enhanced ENDOG-mediated lipid accumulation but had little effect on the lysosome formation (Figure S27a). AKT phosphorylates various downstream substrates, including mTOR, p70S6K, and GSK3 β . Here, we found that AKT overexpression enhanced phosphorylation of GSK3 β and p70S6K both in the ENDOG overexpressed and the control (pLVX3) cells (Figure S27b-c), and the p-p70S6K increased dramatically in the presence of AKT (Figure S27b-c). In contrast, AKT has little effect on the phosphorylation of mTOR, ULK1, and 4EBP1 (Figure S27b-c). Additionally, the repressed lipid accumulation in ENDOG knockout cells was rescued after overexpression of AKT (Figure S27d). While the reduced autophagosome in ENDOG knockout cells has no changes in the presence of AKT (Figure S27d). We also found that AKT mainly increased the p-p70S6K in wild-type and ENDOG knockout cells and gently increased the phosphorylation of GSK3 β but had little influence

on the mTORC1 activation (Figure S27e-f). Besides p70S6K and GSK3 β , reintroducing AKT also increased the phosphorylation of ACLY (Figure 2. o). These data indicated that, at least in our model, AKT promoted lipid accumulation mainly by activating ACLY. Moreover, AKT has little influence on ENDOG-mediated mTORC1 suppression and autophagy.

Figure S27

Our results demonstrated that GSK3 β -mediated phosphorylation of ENDOG enhanced its interaction with 14-3-3 γ , which weakened the binding between 14-3-3 γ with TSC2 and Rictor. The released TSC2 and Rictor repressed mTORC1 or activated mTORC2, respectively. The activation of AKT in ENDOG overexpressed cells promoted the phosphorylation of GSK3 β , which repressed the kinase activity of GSK3 β , acting as a negative feedback loop of ENDOG function.

3. It is also important to study as autophagy in general is associated with decreased lipid accumulation by lipophagy. However, in this case lipogenic pathway can lead to more lipid formation. Additionally, what happened to other lipolytic pathways such as lipolysis by classical lipases when beta-oxidation is not changed (Figure 5A) and autophagy is inhibited (as per the old study)?

Response: As suggested, we performed new experiments to study lipophagy and lipolysis.

Lipophagy was first observed in hepatocytes. Autophagic proteins LC3B and LAMP1 exhibited a close association of lipid droplets when autophagy was inhibited (*Singh R, et al, 2009, Nature, PMID: 19339967*). Here, we found that there are a few colocalizations of lipid droplets (stained with BODIPY) with autophagosome (stained with LC3B) and lysosome (stained with LAMP1) in hepatocytes (Figure S28a-b). Our data found little lipophagy in ENDOG overexpressed or knockout hepatocytes, indicating that autophagy-induced lipid degradation is not the primary manner for decreased lipid accumulation.

Besides lipophagy, lipolysis is a primary kind of lipid droplet degradation. ATGL and HSL are two rate-determining enzymes to catalyze lipolysis. We found that overexpression of ENDOG increased the expression of ATGL and HSL, while loss of ENDOG reduced their expressions (Figure S28c-f). It is possible that increased ATGL/HSL may be caused by the negative feedback of ENDOG-mediated lipid accumulation.

Figure S28

4. Although EndoG KO mice showed less lipid accumulation, what happened to hepatic inflammation in HFD mice when autophagy is also inhibited (as per the old study)? Autophagy has been shown to be key in regulating inflammation during NAFLD.

Response: Chronic low inflammation is considered a marker of NAFLD. In our model, we found that ENDOG knockout repressed HFD-induced NAFLD and HFD-induced inflammation, which was indicated by the reduced transcriptional levels of inflammatory genes and phosphorylation of NF- κ B (data shown below).

Multiple factors trigger liver inflammation, including oxidative stress, lipotoxic, insulin resistance, ER stress, and inhibited autophagy (Petrescu M. et. al, 2022, Medicina

(Kaunas), PMID: 35630058). ER stress plays a critical role in inflammation. IRE1a stimulates inflammation by binding with TRAF2, which promotes the nuclear localization of NF- κ B (Urano, F. et al, 2000, Science, PMID: 10650002). PERK promotes the expression of IL-1 and TNF- α by suppressing I κ Ba (an inhibitor of NF- κ B) (Garg, A.D. et al, 2012, Trends Mol. Med., PMID: 22883813).

Our data showed that loss of ENDOG repressed the expressed IRE1a and PERK both in hepatocytes and mice liver (Figures 5e and 5h-i). Autophagy regulates inflammation in a complex and multiple-dimensional way. Firstly, autophagy regulates the development, homeostasis and survival of inflammatory cells, including macrophages, neutrophils, and lymphocytes. Secondly, autophagy affects the secretion of cytokines. A deficiency of autophagy promotes the secretion of IL-1b (Crisan TO, et al. 2011, PLoS ONE, PMID: 21490934). Thirdly, cytokines also regulate the inducement of autophagy. Th1 cytokines (such as TNF α , IL-1, and IL-6) promote autophagy induction, while, Th2 cytokines (such as IL-4, IL-10, IL-13) inhibit autophagy (Wu TT, et al, 2016, Int J Biol Sci. PMID: 27313501). In the high-fat-diet model, inhibited autophagy was found to promote inflammation and liver fibrosis via lipophagy (lipid droplet degradation) (Gao, J. et al, 2020, J. Hepatol. PMID: 32389810). However, we found that little lipophagy exists in our model and ENDOG has no effects on lipophagy (Figure S28). These data suggested that loss of ENDOG repressed HFD-induced liver inflammation, which may be mainly through the repression of ER stress but not autophagy.

5. There are other transcription factors regulated by ER stress sensors such as PPAR γ , ChREBP, SREBP1c, SREPB2, FABP4, CRT2, CREBH, etc. what happened to these or SREBP1 is the only one affected by Endo G-associated ER Stress? Also, SREBP1c is the predominant isoform in the liver regulating hepatic lipogenesis so including SREBP1c along with the others is more appropriate.

Response: As suggested, we detected the expression of PPAR γ , ChREBP1, FABP4, CRT2, and CREBH in ENDOG knockout livers and cells. As shown below, the mRNA and protein expression of PPAR γ , CHREBP1, FABP4, CRT2, and CREBH were not changed in ENDOG knockout mice, compared with control mice (Figure 18 a-c). Furthermore, we found that neither overexpression nor knockout of ENDOG affects the expression of PPAR γ ,

ChREBP1, FABP4, CRTC2, and CREBH (Figures S18 d-i). These data indicated that SREBP1 is the predominant transcriptional factor in ENDOG-induced lipid synthesis.

Figure S18

6. FASN only partially regulates oleic acid-induced TG synthesis (lipid droplets) and also requires SCD1 activity as oleic acid is a mono-unsaturated fatty acid and not fully dependent on FASN activity (majorly regulating saturate fat palmitic acid formation). Please also include SCD1 and DGATs expression to show if the complete pathway is upregulated.

Response: SCD1 is an ER enzyme critical for the de novo synthesis of monounsaturated fatty acids (J.M. Ntambi, et al, 2004, *Progress in Lipid Research*, PMID: 14654089). DGAT1 and DGAT2 catalyze the final step of triglyceride synthesis (S.B. Weiss, et al, 1960, *J. Biol. Chem.* PMID: 13843753). As suggested, we tested the expression of SCD1, DGAT1, and

DGAT2 in ENDOG knockout livers and cells. Only SCD1 was repressed in ENDOG knockout mice livers and hepatocytes (Figures S14 a-e). Overexpression of ENDOG increased the SCD1 expression in hepatocytes (Figures S14 f-g). However, the expression of DGAT1 and DGAT2 had no changes in ENDOG knockout mice liver/hepatocytes or ENDOG overexpressed hepatocytes (Figure S14). ACC promotes the conversion of acetyl-CoA into malonyl-CoA, and FAS condenses malonyl-CoA and acetyl-CoA to form palmitate. Our data found that loss of ENDOG repressed the expression of ACC, FAS, and SCD1 in mice livers and hepatocytes. These data indicated that ENDOG promotes lipid synthesis by increasing the early processes of lipid synthesis.

Figure S14

7. The authors also showed that insulin treatment restored lipid accumulation, and EndoG activated AKT in the liver. How are the blood glucose and insulin levels in the mice at basal and under starvation? it is important to analyze fasting blood glucose, glucose tolerance, and Insulin tolerance in normal chow-fed and HFD-fed WT and KO/LKO mice. This is especially important to understand if targeting EndoG leads to an improved fatty liver but interferes with other important pathways such as autophagy, insulin resistance, hepatic inflammation etc.

Response: As suggested, we measured the blood glucose and insulin level and performed the GTT and ITT assay. We found that neither on the normal nor HFD chow conditions,

loss of ENDOG did not affect the nonfasting blood glucose in male and female mice (Figures S10 a-d). Liver-specific knockout of ENDOG also had no changes in nonfasting blood glucose in male and female mice after HFD chow (Figures S10. e-f). We also found that loss of ENDOG also did not affect the fasting blood glucose (fasting for 16 hours) in HFD-chow male and female mice (Figures S10 g-h).

Figure S10

Furthermore, we performed the tolerance tests after the high-fat diet for 15 weeks. We found that knockout of ENDOG did not affect glucose tolerance in male and female mice (Figures S12 a-b and e-f). In comparison, the loss of ENDOG increased insulin sensitivity in female mice but not in male mice (Figures S12 c-d and g-h). These data indicated that ENDOG does not affect blood glucose but increases insulin sensitivity in female mice.

We also measured the serum insulin in ENDOG knockout and liver-specific knockout female mice in normal chow, fasting (16 hours), and high-diet conditions. We found that loss of ENDOG does not influence serum insulin (Figures S12 i-j). These data indicated that loss of ENDOG repressed fatty liver probably not through the insulin resistance pathway.

Figure S12

Meanwhile, there is little colocalization between lipid droplets and autophagosome/lysosome in hepatocytes (Figures S28 a-b), indicating that lipophagy has little role in the lipid metabolism of our system. Thus, we believe ENDOG-mediated mTORC2-AKT-ACLY activation and ER stress may be the critical factor for lipid accumulation.

Figure S28

8. The authors need to comprehend their old findings with the new (as both models are the same) once in a way to better understand and link them together.

Response: Our previous work (Nat Commun, 2021) demonstrated that ENDOG promotes autophagy partially via competitively binding with 14-3-3 γ , which released TSC2 to repress mTORC1 activity. In the present study, ENDOG promoted lipid synthesis by activating the mTORC2-AKT-ACLY axis. In these two independent studies, we found that ENDOG bound with 14-3-3 γ , then released two different proteins, TSC2 and Rictor. TSC2 represses mTORC1 by converting RHEB-GTP (an activator of mTORC1) to the inactive form of RHEB-GDP (Brendan D Manning, et al. 2003, Trends Biochem., PMID: 14607085). Rictor, a subunit of mTORC2, has been reported to reduce mTORC2 activity when binding with 14-3-3 (Yeong Ha Jeon, et al. 2013, Biochimica et Biophysica Acta, PMID: 23680186). Besides, it has shown that the TSC1-TSC2 complex is required for proper activation of mTORC2 in a manner independent of its GTPase-activating protein activity toward Rheb (Jingxiang Huang, et al., 2008, Mol Cell Biol., PMID: 18411301; Jingxiang Huang, et al. 2009, Biochem Soc Trans. PMID: 19143635

Thus, in our model (as shown below), we found ENDOG competitively bound with 14-3-3 γ under oleic acid treatment, which released Rictor and TSC2, both of which can activate mTORC2. These data indicated that ENDOG is competitively bound with 14-3-3 γ , which released TSC2 and Rictor and then represses mTORC1 (previous data) or activates

mORC2 (present data), respectively.

9. Lastly, any specific reason for choosing female mice for the NAFLD study as male mice are the preferred choice for such metabolic studies to reduce the hormonal influence? What happens in the male EndoG mice on HFD?

Response: Male mice have lower insulin sensitivity than female mice, which makes male mice more susceptible to high-fat diet-induced obesity and metabolism syndrome (*Wang, et al, 2018, Nat Commun. PMID: 29670083*). We fed both male and female mice with high-fat diets for 15-16 weeks. However, the ENDOG knockout and liver-specific ENDOG knockout did not affect the body weight and nonfasting blood glucose in male mice (Figures S9 a, c and S10 c, e).

Figure S9

Figure S10

The weight of liver, kidney, white fat, and brown fat had no changes in ENDOG knockout and ENDOG liver-specific knockout mice (Figures S9 b and d). Besides, loss of ENDOG

does not influence glucose tolerance and insulin insensitivity in male mice (Figures S12 a-d), while loss of ENDOG reduced body weight and liver and white fat weight in female mice (Figure 3). Thus, we focused more on the role of ENDOG in lipid metabolism in the female mice in the present study. We already pointed out that this function of ENDOG is specific in female mice in the revised manuscript and also discussed it in the discussion section.

Figure S12

The Reviewer believes that the comments will be helpful to increase clarity and improve the undersatding of current work.

Thanks for your advice that would surely help us improve our work.

Reviewer #2 (Remarks to the Author):

In this manuscript, Wang et al. described their observation that ENDOG translocates to the ER, binds with Bip resulting in release of IRE1a/PERK that activates the ER stress response, promoting the expression of SREBP1, ACC, and FAS. Using in vitro and in vivo approaches they have beautifully shown that loss of ENDOG suppresses acetyl-CoA production and lipid synthesis, along with reducing ER stress, which eventually alleviates HFD-induced NAFLD. Although the work of the manuscript is very attractive there are some major concerns.

Major Comments

1. Previous study from Wang et al, has described, "Endonuclease G promotes autophagy by suppressing mTOR signalling and activating the DNA damage response, <https://doi.org/10.1038/s41467-020-20780-2> ."which the reviewer felt a bit contradicting, how do the authors justify two opposite mechanism in once cell type, there need a strong explanation to it.

Response:

Thanks for your question. As suggested, we performed more experiments to explain the confusing point.

Here, in the ENDOG-KO and LKO mice livers, we found that loss of ENDOG activates mTORC1 signaling in both normal and HFD-fed mice, as shown in the increased phosphorylation of ULK1, p70S6K and 4EBP1 (Figure S24a-b and e-f). Besides, we detected a decrease of LC3B-II in ENDOG knockout mice livers, indicating that loss of ENDOG repressed autophagy in normal and HFD-fed mice. Additionally, we found that ENDOG loss repressed the phosphorylation of AKT and ACLY in normal and HFD-fed mice (Figure S24a-b and e-f). These data suggested that loss of ENDOG activated mTORC1 activity and then repressed autophagy in normal and HFD-fed conditions, consistent with our previous work. Meanwhile, loss of ENDOG repressed mTORC2-AKT activation and then suppressed ACLY phosphorylation.

Figure S24

Furthermore, we reproduced our previous data and found that overexpression of GSK3 β promoted ENDOG binds to 14-3-3 γ , which reduced the interaction between 14-3-3 γ with TSC2 and 14-3-3 γ with Rictor. However, the enzyme activity deficient GSK3 β (GSK3 β -Mut) could not affect the binding between ENDOG/Rictor with 14-3-3 γ (Figure S25c). Furthermore, we found that GSK3 β promoted the binding between ENDOG and 14-3-3 γ and reduced the interaction between 14-3-3 γ with TSC2 and Rictor. While, ENDOG-AA (T128S288 to A128A288), which GSK3 β could not phosphorylate, also could not affect the

interaction between ENDOG with 14-3-3 γ , and 14-3-3 γ with TSC2 and Rictor (Figure S25d). These data suggested that GSK3 β -mediated phosphorylation of ENDOG enhanced its interaction with 14-3-3 γ , which weakened the binding between 14-3-3 γ with TSC2 and Rictor. The released TSC2 and Rictor repressed mTORC1 or activated mTORC2, respectively.

Figure S25

Our previous work (Nat Commun, 2021) demonstrated that ENDOG promotes autophagy partially via competitively binding with 14-3-3 γ , which released TSC2 to repress mTORC1 activity. In the present study, ENDOG promoted lipid synthesis by activating the mTORC2-AKT-ACLY axis. In these two independent studies, we found that ENDOG bound with 14-3-3 γ , then released two different proteins, TSC2 and Rictor. TSC2 represses mTORC1 by converting RHEB-GTP (an activator of mTORC1) to the inactive form of RHEB-GDP (Brendan D Manning, et al. 2003, Trends Biochem, PMID: 14607085). Rictor, a subunit of mTORC2, has been reported to reduce mTORC2 activity when binding with 14-3-3 (Yeong Ha Jeon, et al. 2013, Biochimica et Biophysica Acta, PMID: 23680186). Besides, it has shown that the TSC1-TSC2 complex is required for proper activation of mTORC2 in a manner independent of its GTPase-activating protein activity toward Rheb (Jingxiang Huang, et al., 2008, Mol Cell Biol., PMID: 18411301; Jingxiang Huang, et al. 2009, Biochem Soc Trans. PMID: 19143635)

Thus, in our model (as shown down), we found ENDOG competitively bound with 14-3-3 γ under oleic acid treatment, which released Rictor and TSC2, both of which can activate

mTORC2. These data indicated that ENDOG is competitively bound with 14-3-3 γ , which results in the release of TSC2 and Rictor, and then repress mTORC1 (previous data) or activate mTORC2 (present data), respectively.

2. The authors should evenly show the densitometry as some figures have it and some don't, which is a bit of a concern.

Response: We are sorry about not showing all the statistical results of western blots because of the limited figure page. In the revised manuscript, we provided all the statistical results of western blots as suggested.

3. the authors should clearly state the cell lines that has been used and have they validated that cell line or have they checked the reproducibility in any other cell line.

Response: In the present study, we use the HepG2 cell line to study the role of ENDOG in lipid metabolism. We had stated the cell line in the methods part and figure legends in

the revised manuscript. Moreover, we knockdown ENDOG in another three hepatoma cell lines, MHCC97H and HCC-LM3, and Huh7. We found that the knockdown of ENDOG repressed the expression levels of ACC, FAS, Lipin1, and the phosphorylation of AKT and ACLY (Figures S19 a-b, e-f, and i-j). BODIPY staining results also showed that ENDOG loss repressed the oleic acid-induced lipid accumulation (Figures S19 c-d, g-h, and k-l). These data suggested that loss of ENDOG repressed lipid synthesis in several kinds of hepatocytes.

Figure S19

4. Some of the blots looked overexposed, the authors should take a careful look at the blots, supplementary 11 is just an example.

Response: Thanks for this kind suggestion. We replaced all the overexposed WB bands in our revised manuscript. We presented short (S) and long (L) exposed bands for some

proteins that showed precursor and mature bands.

5. The reviewer kept wondering as authors have not clearly mentioned regarding the improvement in the glucose tolerance or insulin levels, it would be good to show that data.

Response: Thanks for your advice. After the high-fat diet of 15-16 weeks, we performed the glucose tolerance tests. We found that knockout of ENDOG did not affect glucose tolerance in male and female mice (Figures S12 a-b and e-f). Loss of ENDOG increased insulin sensitivity in female mice but not in male mice (Figures S12 c-d and g-h). We also measured the serum insulin in ENDOG knockout and live-specific knockout female mice in normal chow, fasting (16 hours), and high-diet conditions. We found that loss of ENDOG does not influence serum insulin (Figures S12 i-j). These data indicated that ENDOG has no effects on blood glucose and has little influence on the insulin sensitivity of female mice.

Figure S12

Reviewer #3 (Remarks to the Author):

The work by Wang, Tan and colleagues investigates the role of Endonuclease G (ENDOG) in liver physiology and its contribution to the development of non-alcoholic fat liver disease (NAFLD). Results illuminate a two-pronged mechanism that together sustains the accumulation of intracellular lipid storage by ENDOG. In fact, ENDOG promotes the de novo production of cytoplasmic acetyl-CoA, a building block for the synthesis of lipids, and induces ER stress to activate SREBP, a meter regulator of lipid biosynthesis. The study builds from prior work on ENDOG and stems from side observations on Endog-ablated cells. The study significantly adds to our knowledge of ENDOG function and is of significant value for multiple research fields being at the intersection between liver pathophysiology, cellular metabolism, signaling. The work is well executed and deploys a number of biochemical and cell biology techniques, as well as multiple Endog-deficient animal models. Experiments are non redundant and results extremely sharp and clear. Ultimately, the data supports authors' conclusions, although proposed mechanisms do have gray areas that could be addressed.

I believe that the robustness of the work might improve if the authors addressed these concerns:

1) the observation that ENDOG-KO reduces LDs is a cornerstone of the manuscript. Given its key position, a better characterization would be appreciated. Can the authors stain for LD markers? Is PLIN2 expression also reduced? Can the authors rule out those bodies are in fact lipid-laden endosomes?

Response: As suggested, we stained the lipid droplet maker PLIN2. We found that in the normal condition, the wild-type cell showed small PLIN2 dots, while loss or knockdown of ENDOG reduced the PLIN2 expressed dots (Figures S1 a-b). Under the oleic acid treatment, the number and size of lipid droplets were dramatically increased (Figures S1 c-d). By co-staining lipid droplets (BODIPY) and PLIN2, we found PLIN2 almost located on the surface of lipid droplets, indicating that the BODIPY positive dots is exactly the lipid droplets but not the lipid-laden endosomes.

Figure S1

2) The manuscript posits that ENDOG-KO limits lipid synthesis and deposition onto LDs. However, the data do not actually rule out the possibility that LDs are smaller because lipids are mobilized and oxidized more. Even if acetyl-CoA levels are augmented, we do not know where that comes from and the directionality of metabolic fluxes. And the result that ACLY depletion decreases LDs upon OA addition might actually indicate that enhanced fatty acid oxidation is in play. I recommend that the authors test the levels of FAO in ENDOG-KO

cells and it would be appreciated if they could perform carbon tracing to determine the source of acetyl-CoA (I acknowledge this might be challenging and beyond the scope of revision)

Response: In our previously submitted manuscript, we found that loss of ENDOG did not affect the mRNA level of several lipid beta-oxidation genes, such as *CPT1*, *CPT2*, and *ACOX1* (Figure 5 a). As suggested, we test two vital rate-limiting enzymes of lipid beta-oxidation, CPT1 and CPT2. Here, we found that neither overexpression nor knockout of ENDOG had no changes on the expression of CPT1 and CPT2 (Figures S15 a-d). We also found that loss of ENDOG did not affect the expression of CPT1 and CPT2 in mice livers (Figures S15 e-f). These data suggested that loss of ENDOG repressed lipid accumulation may not be through lipid oxidation.

Figure S15

Our data found that loss of ENDOG repressed the ACLY activation (phosphorylation of ACLY) both in hepatocytes and mice livers. ACLY is a crucial enzyme that links glucose and lipid metabolism. ACLY catalyzes the glycolytic cascade product citrate to acetyl-CoA, a building block for lipid de novo synthesis. We found ACLY depletion repressed ENDOG-mediated lipid accumulation (Figure S8), indicating that ENDOG promoted lipid accumulation mainly through ACLY-dependent lipid synthesis. Carbon tracing of the acetyl-CoA in ENDOG knockout hepatocytes is an excellent suggestion. We will focus on it in future work.

Figure S8

3) The relationship between ENDOG and ER stress is interesting and significantly adds to existing literature. However, the mechanism could be better characterized. The association of ENDOG with Bip and Gpr75 seems weak and is minimally characterized biochemically and the possibility that ER stress is a secondary consequence of LD decrease has been neglected (e.g.: LDs sequester toxic lipids and alleviate ER stress; PMID: 19081072; 25829424). The manuscript would improve if this possibility is discussed and tested.

Response: As suggested, we performed more experiments to verify the interaction of ENDOG with BIP and Gpr75. In the revised manuscript, we found that BiP could also pull down ENDOG (Figure S21 a). We performed the Co-IP assay again and found ENDOG indeed interacted with BIP and Grp75 (Figure S21 b). Meanwhile, we found that the binding between ENDOG with BiP/Grp75 was increased when treated with oleic acid (Figure S21 c). The ER fraction isolation assay showed that oleic acid treatment promoted the translocation of matured ENDOG from mitochondria to ER (Figures 6 h-i). All these data demonstrated that oleic acid treatment induced the translocation of ENDOG into the ER component, which provided the spatial conditions for interaction between ENDOG and Bip/Grp75.

Figure S21

We agree with you that LDs sequester toxic lipids and alleviate ER stress. However, the relations between ER stress and lipid metabolism are reciprocal and complex. ER stress is a crucial regulator for lipid metabolism and a consequence of lipid overload. ER stress proteins, such as XBP-1, IRE1a, ATF4, PERK, ATF6, and CHOP, regulate the expression of a variety of lipid metabolism enzymes, including lipid synthesis, oxidation, and export (Jaeseok Han, et al, 2016, *J Lipid Res*; PMID: 27146479). An overload of lipids also promotes ER stress, which resulting lipid droplets formation to avoid lipotoxicity (Ivan Hapala, 2011, *Biol Cell*; PMID: 21729000). In D. Thomas's work (PMID: 19081072), they found that loss of ATF6, a sensor of ER stress, promoted lipid deposition in mice livers. They found transcriptional levels of *SREBP1*, *CHREBP1*, *CE/BP1*, *PPARG*, and *PGC1A* were all downregulated in ATF6-null mice, indicating that both lipogenesis and lipid oxidation were repressed when knockout of ATF6. Their systematic investigation found that liver steatosis might be driven entirely by the downregulated lipid oxidation, and the decreased lipid synthesis is negative feedback for liver steatosis.

In our work, we found that loss of ENDOG repressed the expression of ER stress sensors (ATF6, IRE1a, and PERK) both in vitro and in vivo (Figures 5 e and h-k). Consistent with D. Thomas's work, we found that belonging to the decreased ER stress, the lipid synthesis-related proteins (ACC, FAS, and SREBP1) were also repressed in ENDOG knockout mice livers and hepatocytes (Figures 5 e and h-k). While differed from their work, in which they demonstrated reduced lipid oxidation is the key reason for liver steatosis, we found that loss of ENDOG has no changes in lipid oxidation (Figures 5 a and S15). Thus, we think that the reduced lipid synthesis is the major contributor to lipid accumulation when loss of

ENDOGL.

I also have a couple of conceptual concerns on which I would appreciate a comment from authors.

1) acetyl-CoA levels are down in ENDOG-KO cells and mouse livers (Fig 2C-D), and this is ascribed to reduced flux through ACLY. However, a large fraction of the acetyl-CoA pools in hepatocytes is produced from acetate. Have the authors tested the possibility that ENDOG regulates acetate metabolism? or acetate recycling (from de-acetylation reactions)?

Response: Thanks for pointing out this possibility. We detected the expression of ALDH2 and AceCS1, a key enzyme for acetaldehyde converting to acetate and acetate to acetyl-CoA, respectively. We found that neither overexpression nor knockout of ENDOG affected the expression of AceCS1 and ALDH2 in hepatocytes and mice livers (Figure S6). These data suggested that ENDOG has no effects on acetate recycling. The loss of ENDOG-mediated reduction of acetyl-CoA mainly through the repression of ACLY.

Figure S6

2) Cells were supplemented with exogenous acetyl-CoA and rescue in LD and TG content was observed (Fig 2H-J). This puzzles me because acetyl-CoA does not cross the plasma membrane and no transporter is known to my knowledge.

In fact, this is the very first time I see someone attempt a rescue experiment with exogenous supplementation of acetyl-CoA. Can the authors comment on that?

Response: Thanks for pointing out this important question. We agree with you that acetyl-CoA does not cross the plasma membrane. To address this question, we performed more experiments and checked them carefully. Firstly, the exogenous acetyl-CoA we used is Acetyl coenzyme A trisodium salt (Sigma, Cat: A2056-5MG), which may increase its transmembrane activity in an unknown manner. Second, we treated the HepG2 cells with 100 μ M acetyl-CoA (Sigma, Cat: A2056-5MG) and then collected the supernatant medium and cell pellet at different time points for the acetyl-CoA level measurements by ELISA kit. We found that the intracellular acetyl-CoA increased rapidly in the early three to six hours, then maintained at the following time (figure A below). However, we failed to detect the acetyl-CoA in the medium, even at a high concentration of acetyl-CoA trisodium salt (500 μ M, data not shown). While we indeed detected that the intracellular acetyl-CoA was increased after the exogenous treatment of acetyl-CoA sodium. Additionally, by co-staining between lipid droplets (BODIPY) and lipid-protein (PLIN2), we found that exogenous supplementary of acetyl-CoA increased the cell lipid droplets in a dose-dependent manner both in wild-type and ENDOG knockout HepG2 cells (figure B below). These data indicated that extracellular acetyl-CoA could indeed transport into intracellular in some way which we did not know and need further study.

Reviewer #4 (Remarks to the Author):

Zhou and co-workers investigated whether EndoG, a mitochondrial protein, has a role in the cytosol. They found that loss of EndoG decreased acetyl-CoA production, lipid synthesis, and ER stress which eventually relieved HFD-induced NAFLD. This is a high-quality study that provides insights into the HFD-induced NAFLD. Although the observed effects of cytoplasmic EndoG are significant, it remains unclear whether they are influenced or not by the endonuclease activity of the protein. It would be important to determine whether loss of the nuclease activity of EndoG affects acetyl-CoA production, lipid synthesis, and ER stress.

Response: Thanks for your suggestions. To investigate whether the endonuclease activity of ENDOG is necessary for ENDOG-induced lipid synthesis and ER stress, we overexpressed the wild-type ENDOG (ENDOG) and endonuclease activity defective ENDOG (ENDOG-EM) in ENDOG knockout cell. We found that similar to the wild-type ENDOG, overexpression of ENDOG-EM also increased the phosphorylation of AKT and ACLY (below Figure A-B). Our data demonstrate that ENDOG is competitively bound with 14-3-3 γ , which released Rictor to the active mTORC2-AKT-ACLY axis. Here, we found the deficiency of ENDOG's nuclease activity had no influence on the interaction between ENDOG and 14-3-3 γ , and the binding between 14-3-3 γ and Rictor is similar in ENDOG and ENDOG-EM overexpressed cells (figure C below). These data indicated that the enzyme activity of ENDOG is not necessary for its interaction with 14-3-3 γ , and the following Rictor release, mTORC2-AKT-ACLY activation. On the other hand, our data demonstrate ENDOG translocated to ER component and bound with Bip to promote ER stress, which increased the expression of SREBP1, FAS, and ACC. We found that both ENDOG and ENDOG-EM increased the expression of IRE1a and XBP-1, SREBP1, FAS, and ACC (figure D-E below). The binding of Bip with ENDOG has no changes when there is a loss of enzyme activity (figure F below). These data show that ENDOG interacted with Bip, prompted ER stress, and the expression of lipid synthesis proteins is endonuclease activity independent.

The BODIPY staining and triglyceride measurement results also demonstrated that ENDOG and ENDOG-EM could promote lipid accumulation under the oleic acid treatment

(figure G- H below). More importantly, we used the ENDOG-specific inhibitors PNR-3-80 and PNR-3-82, which repressed the endonuclease activity of ENDOG, to block ENDOG's enzyme activity. ENDOG inhibitor treatment could not repress oleic acid-induced lipid accumulation (figure I-J below). These data indicated that the endonuclease activity of ENDOG is not necessary for ENDOG-mediated ER stress and lipid synthesis.

Comments:

Fig. 2A-B:

The effects of EndoG on AKT phosphorylations appear to be quite modest. It would be important to quantify the relative intensities of the bands observed in the Western blots against AKT S473 and AKT T308.

Response: As suggested, we quantify the relative intensities of all the western blots.

Fig. 5E:

Treatment of WT cells with TM appears to increase the PERK signal only weakly. Could you quantify the intensities of the bands observed in the Western blots against PERK in the experiments involving the TM and TG-treated cells?

Response: As suggested, we quantify the WB of Figure 5e, as shown in Figure S16 b. We found that TG treatment increased IRE1a, ATF6, and PERK. While TM treatment only increased the ATF6 weakly. However, the biomarker of ER stress, CHOP, and XBP-1 increased in TG and TM treatment.

Figure S16

REVIEWERS' COMMENTS

Reviewer #1 (Remarks to the Author):

The Reviewer acknowledges the authors' diligent efforts in conducting essential experiments to enhance the clarity and scientific robustness of this manuscript.

The revised version is deemed satisfactory, with no further comments.

Reviewer #2 confidentially expresses support for the manuscript.

Reviewer #3 (Remarks to the Author):

Congratulations to all authors for the terrific job. The manuscript is significantly improved and can certainly be published in Nat Comm

Reviewer #4 (Remarks to the Author):

The revised manuscript by Zhou and co-workers provides insights into the role of ENDOG in the cytosol. The authors constructively responded to my criticism and the revision improved the manuscript. It would be important to include in the revised manuscript the new data that showed that the loss of the endonuclease activity of ENDOG did not affect the ability of the protein to influence the ER stress and lipid synthesis.